# ATLAS 🌐: Adaptive Transfer Scaling Laws for Multilingual Pretraining, Finetuning, and Decoding the Curse of Multilinguality

**Shayne Longpre**[*,1] **Sneha Kudugunta**[2] **Niklas Muennighoff**[3] **I-Hung Hsu**[4]
**Isaac Caswell**[5] **Alex Pentland**[1] **Sercan Ö. Arık**[4] **Chen-Yu Lee**[4] **Sayna Ebrahimi**[5]

[1]MIT  [2]University of Washington  [3]Stanford University  [4]Google Cloud AI  [5]Google DeepMind

## ABSTRACT

Scaling laws research has focused overwhelmingly on English—yet the most prominent AI models explicitly serve billions of international users. In this work, we undertake the largest multilingual scaling laws study to date, totaling 774 multilingual training experiments, spanning 10M-8B model parameters, 400+ training languages and 48 evaluation languages. We introduce the ADAPTIVE TRANSFER SCALING LAW (ATLAS) for both monolingual and multilingual pretraining, which outperforms existing scaling laws' out-of-sample generalization often by more than $0.3\ R^2$. Our analyses of the experiments shed light on multilingual learning dynamics, transfer properties between languages, and the curse of multilinguality. First, we derive a cross-lingual transfer matrix, empirically measuring mutual benefit scores between $38 \times 38 = 1444$ language pairs. Second, we derive a language-agnostic scaling law that reveals how to optimally scale model size and data when adding languages without sacrificing performance. Third, we identify the computational crossover points for when to pretrain from scratch versus finetune from multilingual checkpoints. We hope these findings provide the scientific foundation for democratizing scaling laws across languages, and enable practitioners to efficiently scale models—beyond English-first AI.

## 1 INTRODUCTION

Scaling laws research has focused overwhelmingly on English (Kaplan et al., 2020; Hoffmann et al., 2022; Li et al., 2025a). However, most of the major closed and open models now explicitly target massively multilingual uses (OpenAI, 2025; Anthropic, 2025; Google DeepMind, 2025; DeepSeek-AI, 2024; Team OLMo et al., 2024). Nonetheless, multilingual scaling laws research in the public sphere is limited. Among proprietary lab reports, only Llama-3 briefly discuss their multilingual scaling laws, though they train on only 8% non-English tokens (Dubey et al., 2024). Prominent public work investigates scaling laws for data mixing (Goyal et al., 2024; Ye et al., 2024; Ge et al., 2024a), scaling laws for machine translation (Fernandes et al., 2023; Gordon et al., 2021), multilingual instruction tuning/adaptation (Shaham et al., 2024; Lai et al., 2024; Weber et al., 2024), and only a couple of recent rigorous contributions investigate scaling for smaller multilingual models ($<45M$ and $<1.2B$ respectively) (Chang et al., 2024; He et al., 2024).

Extending these prior works, we seek to fill the ample remaining knowledge gaps with comprehensive examinations of the following research questions. First, we explore how the properties of different languages' scaling laws differ. Second, we measure the cross-lingual transfer benefits between 38 languages. To our knowledge, this represents the most comprehensive available empirical resource to explicitly measure language transfer across $38 \times 38 = 1444$ language pairs. Third, we succeed in modeling the *curse of multilinguality*—the phenomena where adding languages to the training mixture can degrade loss for each language, due to limited model capacity. Lastly, we measure when it is more efficient to pretrain from scratch, or finetune from a general-purpose multilingual checkpoint, resolving an unaddressed practical question. In pursuing these research ques-

---

*Correspondence: slongpre@media.mit.edu

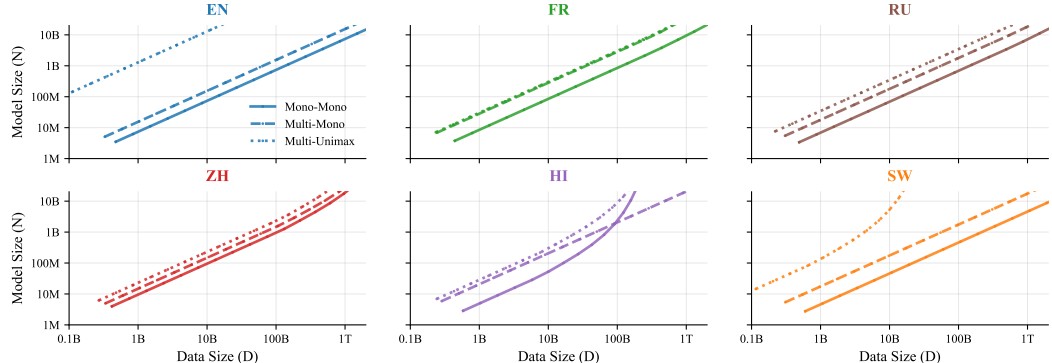

Figure 1: **Optimal Scaling Trajectories** for English, French, Russian, Chinese, Hindi, and Swahili. The law for [monolingual vocabulary, monolingual training]=(—), the law for [multilingual vocabulary, monolingual training]=(- - -), and the law for [multilingual vocabulary, unimax training]=(⋯). **We find (1) per-language optimal scaling trajectories are similar, (2) there is a compute efficiency tax for training with *multilingual* vocabularies or training sets (especially for English), and (3) as Hindi and Swahili observe data repetition their curves slope upward from diminishing returns.**

tions, we develop new tools and estimates for multilingual practitioners, as well as the ADAPTIVE TRANSFER SCALING LAW, which significantly outperforms prior work across several dimensions of generalization. In total, our pretraining and finetuning experiments span 774 independent experiments on MADLAD-400 (Kudugunta et al., 2024), for models sized $10M - 2B$, and evaluating 48 languages. Our main contributions include:

1. **The ADAPTIVE TRANSFER SCALING LAW** that offers better multilingual generalization to larger/unseen $N, D, C$, and training mixes $M$, than prior work (Table 1). In Figure 1 we also estimate a compute efficiency tax between multilingual and monolingual training.

2. **A $38 \times 38$ CROSS-LINGUAL TRANSFER MATRIX**, providing the largest resource for empirically measured transfer benefits/interference between languages (Figure 2).

3. **A scaling law for the *curse of multilinguality***, that informs practitioners how much to scale $(N, D)$ to accommodate expansions in a models' language coverage (Figure 5).

4. **A general pretrain vs finetune formula**, that informs practitioners whether it is more efficient to pretrain from scratch or begin from a multilingual Unimax checkpoint (Figure 7).

## 2 EXPERIMENTAL SETUP

**Datasets and Evaluation** We use the MADLAD-400 dataset (Kudugunta et al., 2024), a popular CommonCrawl-based dataset with the most expansive coverage of languages, totaling over 400. The MADLAD-400 authors prioritized multilingual-specific curation for this pretraining corpus, including language detection, filtering, and preprocessing. For 50 languages, chosen to represent a range of language families, scripts, and degrees of resourcefulness, we partition a random test set, to evaluate vocabulary-insensitive loss (Tao et al., 2024) for fairer cross-lingual comparisons. Following (Muennighoff et al., 2024; He et al., 2024), we evaluate the fit of our parametric scaling laws using $R^2$ calculated on held-out test sets partitioned to assess specific dimensions of generalization: $R^2(N)$ for the largest model sizes, $R^2(D)$ for the largest token ranges, $R^2(C)$ for the largest compute runs, and $R^2(M)$ for unique training language mixtures not seen at training (more details in Appendix B.3). We separate these dimensions in response to prior work that has demonstrated the importance of rigorous test-set splits in scaling laws research (Li et al., 2025a).

**Model Training** We pretrain models from 10M to 8B parameters, using hyperparameter choices similar to those trained in Kudugunta et al. (2024) (with updates as prescribed by recent work). In this work, we train both monolingual and multilingual models, varying the model size $N$, training tokens $D$, multilingual data proportions, and the total training compute $C$. In particular, we focus on

training monolingual models, bilingual models, and massively multilingual Unimax models (Chung et al., 2023). most experiments center around scale, token, and mixture variations on these languages: English, French, Chinese, Hindi, Swahili, Russian, Spanish, Portuguese, Japanese, though certain experiments evaluate across 50 languages. We use a 64k Sentence Piece Model vocabulary (Kudo, 2018). Across experiments we pretrain 280 monolingual models, 240 bilingual models, 120 multilingual mixtures, and we finetune 130 monolingual models from Unimax checkpoints, totaling over 750 independent training runs. Full experimental details, including the specific language choices, hyperparameters, and training mixtures are provided in full in Appendix B.

Table 1: The $R^2$ evaluation metrics for the fitted scaling laws, holding out separate dimensions of generalization: the largest model sizes $N$, most training tokens $D$, most compute $C$, and unique multilingual training mixtures $M$. In both monolingual and multilingual settings, we average $R^2$ across languages=[EN, FR, RU, ZH, HI, SW], including [ES, DE] for the multilingual setting. **We find ADAPTIVE TRANSFER SCALING LAW outperforms prior work in Monolingual and Multilingual settings.** We ablate the use of the terms for $D_{\text{other}}$ and transfer languages ($\sum_{\mathcal{K}_t} D_i$).

| | SCALING LAWS | $R^2$ | $R^2(N)$ | $R^2(D)$ | $R^2(C)$ | $R^2(M)$ |
|---|---|---|---|---|---|---|
| MONO. | CHINCHILLA SCALING LAW (Hoffmann et al., 2022) | 0.94 | 0.68 | 0.94 | 0.90 | – |
| | DATA-CONSTRAINED SCALING LAW (Muennighoff et al., 2024) | 0.93 | 0.78 | 0.93 | 0.88 | – |
| | [Ours] ATLAS ($D_t$ ONLY) | 0.92 | **0.88** | 0.91 | 0.88 | – |
| MULTI. | CHINCHILLA SCALING LAW (Hoffmann et al., 2022) | 0.64 | -0.99 | 0.72 | 0.66 | 0.61 |
| | MULTILINGUAL SCALING LAW (He et al., 2024) | 0.67 | -0.65 | 0.73 | 0.67 | 0.70 |
| | [Ours] ATLAS ($D_t$ ONLY) | 0.70 | -0.75 | 0.80 | 0.72 | 0.64 |
| | [Ours] ATLAS ($D_t + D_{\text{other}}$) | **0.98** | **0.89** | **0.97** | **0.97** | 0.66 |
| | [Ours] ATLAS ($D_t + D_{\text{other}} + \sum_{\mathcal{K}_t} D_i$) | **0.98** | **0.89** | 0.96 | **0.98** | **0.82** |

# 3 ADAPTIVE SCALING LAWS FOR MONOLINGUAL & MULTILINGUAL SETTINGS

**Challenges with existing scaling laws for multilingual modeling.** In this section, we discuss our attempts to precisely model different monolingual and multilingual pretraining experiments. Scaling laws for English are typically based on Chinchilla (Hoffmann et al., 2022)—which we refer to as the CHINCHILLA SCALING LAW (CSL). Hoffmann et al. (2022)'s two power-law terms (Equation (1)), allows us to separate how loss scales model size $N$ and data size $D$. $E$ is the irreducible loss (representing the entropy of natural text), $(A, B)$ are learned coefficients and and $(\alpha, \beta)$ are scaling exponents for for model size and data size respectively. Beyond English, languages may have different scaling laws (Arnett & Bergen, 2025), and language resources are often limited, requiring a DATA-CONSTRAINED SCALING LAW (DCSL) (Muennighoff et al., 2024) to account for diminishing returns after multiple epochs. However, DCSL requires ample data both before and after one epoch to accommodate its two-stage fitting process. For high-resource languages (English, French, Chinese), it is costly to collect ample data beyond one epoch, and for low/mid-resource languages (even Hindi, Hebrew, and Swahili) it can be very difficult to collect sufficient observations below one epoch.[1] As such, even for monolingual modeling, we require a scaling law that is data repetition-aware, and robust to different collection allowances for $(N, D)$.

For modeling multilingual mixtures, monolingual scaling laws can be used (either with $D$ representing the total or target language data), or He et al. (2024) introduce MULTILINGUAL SCALING LAW (MSL), which expresses the loss for target language $t$ using $N, D$ and the sampling ratio for the target languages' language family in the training mixture. However, each of these solutions only accounts for one of: the sampled target language data $D_t$, the total data altogether $D_t + D_{\text{other}}$, or a set of the target and likely positive transfer languages $D_t + \sum_i D_i$. We posit that a better model would separate and learn to weight these independent contributions. In summary, none of the existing multilingual solutions account for (a) multi-epoch data repetition, or (b) the cross-lingual transfer effects beyond the target's language family.

---

[1] In MADLAD-400 these languages have $< 0.3\%$ English tokens. At standard batch sizes, even frequently evaluating (every $1k$) isn't enough to collect many (sometimes any) observations before 1 epoch is reached.

$$\mathcal{L}(N, \mathcal{D}_{\text{eff}}) = E + \frac{A}{N^{\alpha}} + \frac{B}{\mathcal{D}_{\text{eff}}^{\beta}} \tag{1}$$

$$\mathcal{D}_{\text{eff}} = \underbrace{\mathcal{S}_{\lambda_t}(D_t; U_t)}_{\text{Monolingual}} + \underbrace{\sum_{i \in \mathcal{K}} \tau_i \, \mathcal{S}_{\lambda_i}(D_i; U_i)}_{\text{Transfer Languages}} + \underbrace{\tau_{\text{other}} \, \mathcal{S}_{\lambda_{\text{other}}}(D_{\text{other}}; U_{\text{other}})}_{\text{Other Languages}} \tag{2}$$

$$\mathcal{S}_{\lambda}(D; U) = \begin{cases} D, & D \leq U \quad (\leq 1 \text{ epoch}) \\ U\left[1 + \frac{1 - \exp(-\lambda(D/U - 1))}{\lambda}\right], & D > U \quad (> 1 \text{ epoch}) \end{cases} \tag{3}$$

**The ADAPTIVE TRANSFER SCALING LAW.** To resolve these challenges, we introduce the ADAP-TIVE TRANSFER SCALING LAW (ATLAS), a simpler variant of DCSL that is repetition-aware, introduces fewer additional parameters (for the monolingual variant), is fit in one stage, and is adaptable to an open-ended number of languages that would benefit from an explicit cross-lingual term. In Equation (1) the core scaling law formula includes the standard parameters governing the irreducible loss and $N, D$ scaling: $E, A, B, \alpha, \beta$. Its effective data exposure term $D_{\text{eff}}$ (Equation (2)) unpacks data sources into three terms: a Monolingual term for the target language data $D_t$, an optional Transfer Language term for up to $|\mathcal{K}_t|$ languages we wish to learn independent transfer coefficients for $\tau_i$, and an Other Languages remainder term that sums all training tokens not already accounted for in the first two terms $D_{\text{other}} = D_{\text{tot}} - D_t - \sum_{\mathcal{K}_t} D_i$. The latter two terms are optionally added for multilingual modeling, and $\mathcal{K}_t$ can be adapted as any combination of the languages with pre-sumed highest positive transfer to the target language, or languages with the greatest representation in the experiment training mixture. We found the latter was most effective and selected the $|\mathcal{K}_t| = 3$ most highly co-sampled languages along target language $t$ across mixtures. For each term, we apply a saturation function (Equation (3)), ensuring a smooth decay on the effective data for each subsequent epoch where the training tokens have surpassed that languages' unique tokens $U$. The new parameters introduced over standard scaling laws are: the repetition parameter $\lambda$, which is shared across each data source, the transfer weights $\tau_i$ for each language in $\mathcal{K}_t$ (which are initialized from language transfer scores derived in Section 4), and the transfer weight $\tau_{\text{other}}$ for the remainder of language tokens.

**Research Question:** *How do scaling laws differ by language, and by monolingual vs multilingual training mixtures?*

**Monolingual scaling behavior is consistent in form, and multilingual variants exhibit a variable-sized compute-efficiency tax.** Figure 1 compares optimal scaling trajectories across six-variable resourced languages, under three regimes: *monolingual vocabulary + monolingual training*, *multilingual vocabulary + monolingual training*, and *multilingual vocabulary + unimax training*. Across languages, the monolingual curves are nearly parallel, indicating similar exponents and comparable returns to additional data and parameters (see full law parameters in Table C.1). Both multilingual vocabulary and Unimax training shift the frontier upwards, evidencing a *compute-efficiency tax* relative to the monolingual-vocabulary/monolingual-training setting. This is most pronounced for English, indicating it benefits less from language transfer than other languages do from English. For Hindi and Swahili, the right tail bends upward, consistent with diminishing returns from severe data repetition after many epochs. These qualitative trends motivate a single functional form that remains stable across languages while accounting for vocabulary/training-regime effects.

**Research Question:** *How well can our scaling laws capture unique monolingual constraints, and complex multilingual cross-lingual transfer dynamics?*

**ATLAS outperforms prior scaling laws, with a more robust fit across held-out axes in monolingual and multilingual settings.** Table 1 evaluates fitted laws by holding out the largest model sizes $N$, token ranges $D$, compute $C = 6ND$, and held-out language mixtures $M$, reporting $R^2$ averaged over available languages. In the monolingual setting, all scaling laws perform comparably across most dimensions of generalization. However, when generalizing to the largest size of model, laws that model data repetition outperform those that don't. ATLAS provides the strongest generalization to greater model sizes: $R^2(N) = \mathbf{0.88}$ versus $0.68$ (CSL) and $0.78$ (DCSL), respectively.

In the multilingual setting, monolingual scaling laws can only observe their target data tokens, and as such perform adequately, but not well. In particular, they are unable to generalize to the largest

models $R^2(N)$ at all. This is also the case for MSL, although it obtains better generalization to unseen language mixture in $R^2(M) = 0.69 > 0.64$ by virtue of modeling the whole language family. By separating the sources of data, ATLAS is able to significantly outperform the other laws across all dimensions, frequently achieving $> 0.9$ fit. However, our ablation of the data terms shows that only by adding the Transfer Language term is the model able to outperform MSL and achieve a better multilingual mixture generalization $R^2(M) = 0.82$. In summary, ATLAS offers a simple framework to adaptively model cross-lingual transfer, and enables reliable extrapolation across out-of-sample dimensions in both monolingual and multilingual settings—addressing a critical gap where existing approaches fall short.

# 4 How Do Languages Benefit or Interfere with Each Other?

**Research Question:** *Which languages synergize or interfere most with one another's performance?*

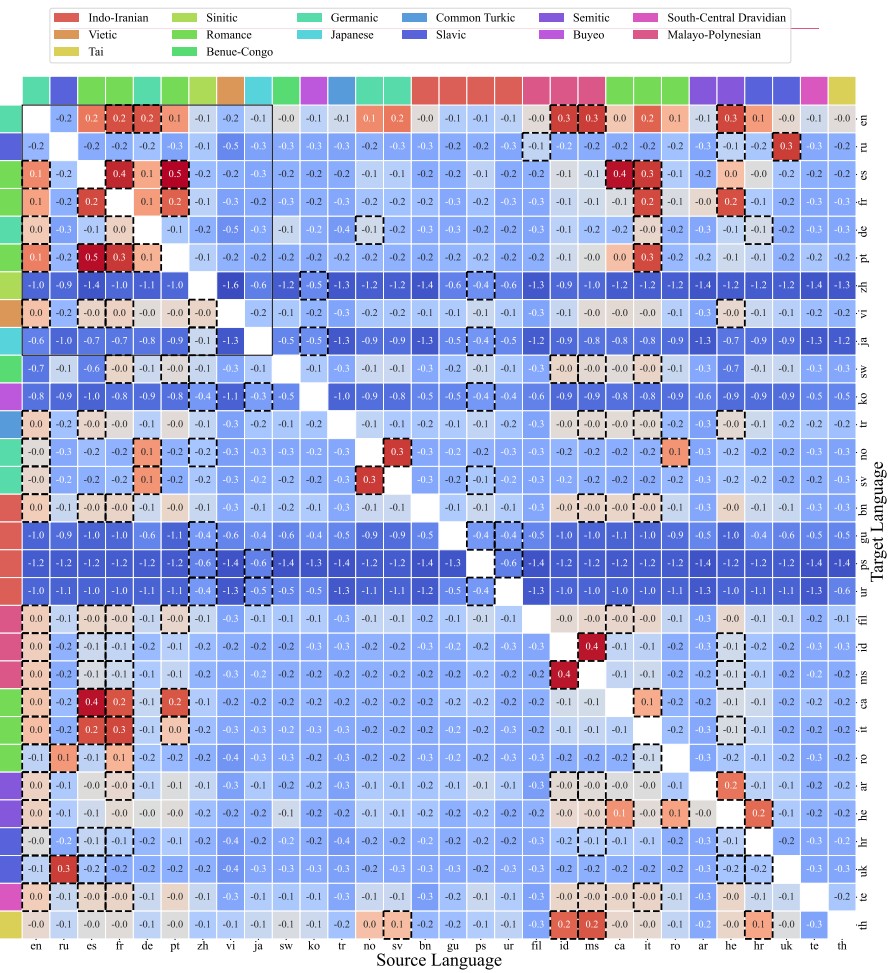

Figure 2: The CROSS-LINGUAL TRANSFER MATRIX, depicting the measured Language Transfer Score across $30 \times 30$ language pairs. Positive scores indicate more positive transfer, negative scores more interference, during bilingual co-training. The dashed boxes indicate the top-5 source languages for each target language. Refer to Appendix B.6 for full details, and Figure C.2 for the larger $38 \times 38$ matrix. **While English is the best source language for many of the languages, we find language similarity is highly predictive of these scores.**

When allocating multilingual training mixtures, practitioners need to know which *source* language(s) provide the most positive benefit, in co-training, to the *target* language(s) they are optimizing for. Throughout this paper, we use the terms transfer or interference to refer to the positive

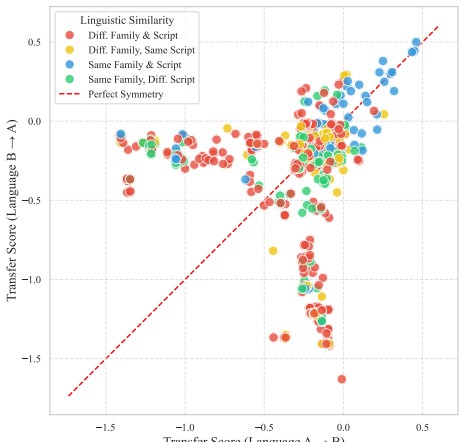 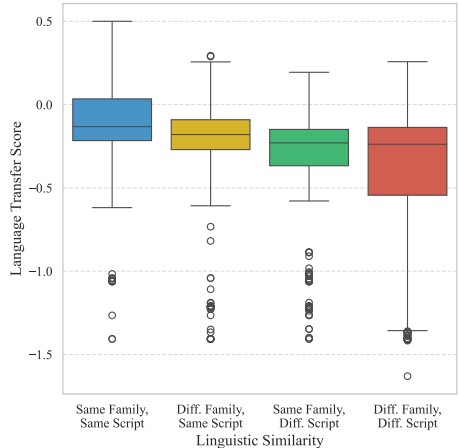

Figure 3: **Left:** A language transfer scatter plot, comparing score symmetry: is language A as helpful to B as vice versa? Points cluster near the diagonal, indicating strong symmetry. **The most synergistic and symmetric pairs almost exclusively share both language family and script.** Greater linguistic distance correlates with increased asymmetry and reduced positive transfer. **Right:** The impact of linguistic similarity (via language family or script) on transfer scores. The box spans the inter-quartile range, with the median line at the center. **We find the differences between each group are statistically significant ($p < .001$), suggesting that sharing either a language family or a script independently contribute greater positive cross-lingual transfer.**

(or negative) inductive transfer between the training signal of two or more tasks (Caruana, 1997). We construct, to our knowledge, the most comprehensive matrix of language-to-language transfer scores, to assist practitioners in selecting training languages.

$$\text{Language Transfer Score } (s \to t) = \text{BTS}_{s \to t} = -\frac{\sigma_{\text{bi}}(L_{\text{t}}(d_{\text{mono}})) - 2d_{\text{mono}}}{d_{\text{mono}}}$$

To do this, we measure how much training in a source language $s$ affects the test loss on a target language $t$. We define a Bilingual Transfer Score (BTS) as the relative training efficiency of a bilingual model $(s, t) = (50\%, 50\%)$ compared to the monolingual model $t$ at reaching the same loss level.[2] $d_{\text{mono}}$ denotes a pre-defined target step (42B tokens), $L_t(d)$ computes the monolingual models' loss at step $d$, and $\sigma_{\text{bi}}$ finds the number of tokens for the bilingual model to reach that loss. Because $\text{BTS}_{s \to t} > 0$ is the zero-centered, when it is $= 0$ there is no transfer, $> 0$ there is positive transfer, and $< 0$ there is negative interference, leading to the bilingual model taking more than double the steps $d_{mono}$ the monolingual model took to reach the target loss $L_{\text{t}}(d_{\text{mono}})$. Refer to Appendix B.5 for full experimental details and methodology.

In Figure 2, we plot the normalized BTS scores between $30 \times 30$ language pairs (or the full $38 \times 38$ in Figure C.2), spanning language families and scripts, on 2B parameter models. Where prior work has developed comprehensive resources for measuring syntactic/phonological distances between languages (Khan et al., 2025), or measured transfer scores with smaller models specifically between high and low-resource languages (Protasov et al., 2024), our work offers among the most expansive and rigorous, fully-symmetric transfer matrices, to our knowledge. It contains many compelling observations. For instance, as one might expect, notable positive transfer exists between related languages, such as between Spanish, Catalan, and Portuguese, or Swedish and Norwegian, or Indonesian and Malay. The top-5 most helpful source languages to co-train with per target are highlighted, with English appearing the most (19 of 30 instances), followed by French (16 of 30), Spanish (13 of 30), and even Hebrew (11 of 30). In other cases, low-resource languages such as Urdu or Pashto have significant negative transfer with all other languages considered.

---

[2]We measure $\text{BTS}_{s \to t}$ directly for 80 language pairs, then estimate it using other training signals (with high fidelity $R^2 = 0.85$), as detailed in Appendix B.6.

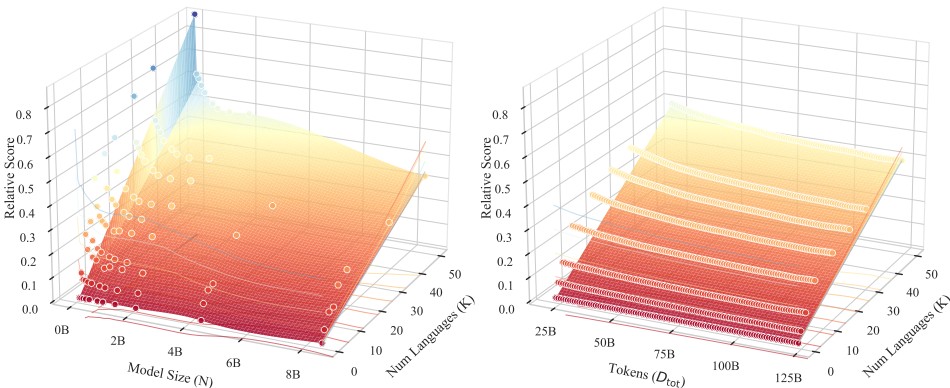

Figure 4: We empirically measure the relative degradation in loss (y-axis), as compared to a monolingual model of the same $(N, D)$, from adding pretraining languages (z-axis). **Left:** We fix $D = 25B$ tokens, and vary the model size $N$. **Right:** We fix $N = 2B$ parameters, and vary the tokens $D$. The points are real empirical observations, averaged across languages. The mesh-grid is a surface estimate, using a cubic spline. **Target language loss is most affected by the number of training languages, but this loss penalty declines for larger models with more capacity.**

**Language script, followed by language family are strongly correlated with positive transfer scores.** In Figure 3, we correlate the transfer scores from the full language matrix Figure 2. Our analysis confirms a clear and statistically significant link between transfer performance and the similarity of the source and target language. Specifically, we found sharing a script or family introduced statistically significant shifts in the mean of the transfer score (always $p < .001$). These findings corroborate (He et al., 2024)'s scaling laws that grouped data by language family. This effect is strongest for shared scripts. As shown in Figure 3, language pairs that share the same writing system (e.g. Latin) exhibit dramatically improved transfer scores compared to pairs with disparate scripts (e.g., Latin and Cyrillic), with a mean score of $-0.23$ versus $-0.39$. The larger effect size for script suggests that the ability to share surface-level representations and subword vocabularies is a primary mechanism for positive transfer, more so than deeper grammatical or lexical similarities alone. In Section 5 and Appendix C.2 we explore how $N, D$ affect language transfer.

**Language transfer scores are often symmetric within the same language family and script, but surprisingly, they cannot be assumed to be reciprocal otherwise.** Next, we examine the extent to which language transfer is symmetric, that is, whether the benefit from language A to B is correlated to that from B to A. As illustrated in Figure 3 (left), we find a surprising triangle pattern, with a Pearson correlation of $r = -0.11$. This implies two things. First, and perhaps most importantly, across all language pairs, there is no significant correlation, and because we observe language A is helpful to language B, *we cannot assume the same in reverse.* This corroborates findings in Li et al. (2025b) that altruistic languages don't always yield mutual benefits. Second, there does appear to be a clear structure to the scatter: language pairs that share family and script (blue) cluster mostly in the top right quadrant and more tightly around the identity line, whereas language pairs with differing script and family (red) are simultaneously less symmetric, and less synergistic. For instance, pairs (French, Spanish) and (Russian, Ukrainian) share highly symmetric transfer scores, whereas (Chinese, Farsi) and (Russian, Vietnamese) are highly asymmetric. We hope these findings, as well as the Figure 2 transfer matrix, will be directly applicable to practitioners selecting language mixtures based on empirical measurements, rather than intuition.

## 5 THE CURSE OF MULTILINGUALITY—MODEL CAPACITY CONSTRAINTS

**Research Question:** *Is the "curse of multilinguality" measurable—the phenomenon where adding languages to the training mixture can degrade loss of each language, due to limited model capacity?*

A model's capacity can hinder its ability to learn multiple languages at once—known as the *curse of multilinguality* (Conneau et al., 2019; Chang et al., 2024). We can empirically measure this relationship, between $N$, $D$, and the number of training languages $K$, by running experiments with variable $K$. Full experimental details and scaling law derivations are provided in Appendix B.7.

**Modeling the Relationship between $K$, $N$, and $D$.** In Figure 4 we examine the relative loss increase, over a monolingual baseline, from varying the number of training languages $K$, and either the model size $N$ (left plot) or total training tokens $D_{\text{tot}}$ (right plot). For simplicity, we don't explicitly model token repetition, and we sample tokens from each language uniformly—we believe this is a reasonable assumption for models designed to serve all $K$ languages. We observe three phenomena. First, the number of training languages $K$ has far more impact on the relative loss than $N$ or $D$. Relative loss appears to grow monotonically as $K$ increases. Second, both increasing model size $N$ and total tokens $D_{\text{tot}}$ mitigate this penalty. However, computing the partial along the fitted surface shows $|\partial S/\partial \log N| > |\partial S/\partial \log D|$; indicating $N$ delivers larger marginal gains than $D_{\text{tot}}$. Consequently, maintaining performance as $K$ grows requires scaling both data and model size, but scaling the model size has a greater effect. To understand the relationship among these variables more precisely, we model per–target–language loss as

$$L(K, N, D_t) \;=\; L_\infty \;+\; A\,\frac{K^\phi}{N^\alpha} \;+\; B\,\frac{K^\psi}{D_t^\beta},$$

where, under even sampling the total tokens across languages are $D_{\text{tot}} = K\,D_t$. We tested several scaling law variations for this research question, but settled on this one as (a) it retains Chinchilla-style power-law decay properties that cleanly separate model capacity from data (it also reduces to Chinchilla when K=1), (b) it makes the role of the number of languages explicit and interpretable, and (c) it achieved a robust $R^2 \geq 0.87$, indicating a strong fit. The exponent $\phi$ captures how capacity requirements grow with the number of languages, while $\psi$ captures how data requirements change with multilinguality. $\psi < 0$ indicates *positive transfer* (sublinear data needs per language as $K$ increases), and $\psi > 0$ implies negative transfer/interference.

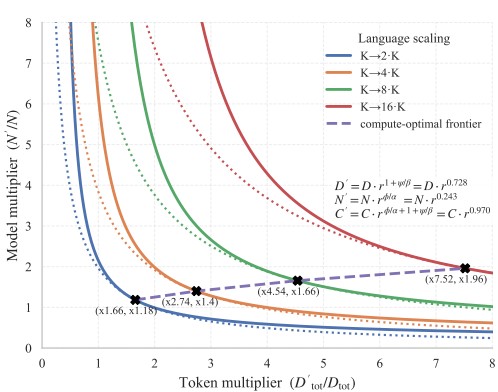

We fit the scaling law for each of 8 different languages, as well as the combination of all, achieving similar coefficients, and $> 0.8$ $R^2$ for each. Using the scaling law fit on all language data we find $\phi = 0.11$ and $\psi = -0.04$, indicating a mild capacity-driven curse of multilinguality, tempered by positive transfer across languages. In other words, as languages are added to the training mixture, a positive $\phi$ means the loss will decline from limited model capacity, however, a negative $\psi$ means that less data per language is needed. (Though the total amount of data increases, as we sample from new languages too.) This provides clear, measurable evidence of positive cross-lingual transfer.

Figure 5: **Iso-loss from expanding language coverage:** When scaling the number of languages a model serves from $K \to r{\cdot}K$, we can estimate how much they need to increase model size $N'/N$ and/or training tokens $D'_{tot}/D_{tot}$, without degrading any of their languages' loss. We derive the compute optimal scaling equations. **The iso-losses indicate that a practitioner should expand their compute budget by $C \cdot r^{0.97}$ to expand their language coverage by $r$.**

**Estimating the Iso-Loss Frontier, and Compute-Optimal scaling of $N, D$ when adding languages:** $K \to rK$. Next, we examine the practical case where a practitioner wants to retrain a new model, increasing the number of languages it serves from $K$ to $rK$, without hindering the performance the original model achieved on the existing languages. To accommodate these new languages, they will need to scale $C$ with some combination of $N \to N'$ and $D_{\text{tot}} \to D'_{\text{tot}}$. We can compute this iso-loss by equating the loss terms: $L(K, N, D_{\text{tot}}) = \frac{AK^\phi}{N^\alpha} + \frac{BK^\psi}{D^\beta} = \frac{A(Kr)^\phi}{N'^\alpha} + \frac{B(Kr)^\psi}{D'^\beta}$. From this we can derive the closed-form expression for how to scale $(N, D_{\text{tot}})$ when increasing $K$ to $rK$

(as derived in Appendix B.7):

$$\frac{N^*(rK)}{N^*(K)} \;=\; r^{\phi/\alpha}, \qquad \frac{D_t^*(rK)}{D_t^*(K)} \;=\; r^{\psi/\beta}, \qquad \frac{D_{\text{tot}}^*(rK)}{D_{\text{tot}}^*(K)} \;=\; r^{1+\psi/\beta}.$$

We can also derive the minimal increase in the compute budget:

$$\frac{C'}{C} \;=\; \frac{N'D'_{tot}}{ND_{tot}} \;=\; \left(\frac{N'}{N}\right)^{\star}\left(\frac{D'_{\text{tot}}}{D_{\text{tot}}}\right)^{\star} = r^{1+\phi/\alpha+\psi/\beta}.$$

In Figure 5 we plot the iso-loss frontiers, and compute-optimal allocations, when scaling a model from $K$ to $r \cdot K$ languages. It shows in closed-form how a practitioner should scale $D_{tot}$, $N$, and by extension their compute budget $C$ to accommodate additional languages, without degrading their loss. For instance, we find expanding to $4 \cdot K$ languages requires a practitioner to expand $D_{tot}$ by 2.74, and the model size $N$ by 1.4. Incidentally, evenly sampling across languages, this actually corresponds to using sampling $1 - (2.74/4) = 32\%$ less data per language. Although there are fewer tokens in each target language (by 32%), the total tokens has risen by 2.74, and the positive transfer prevents any potential loss degradation.

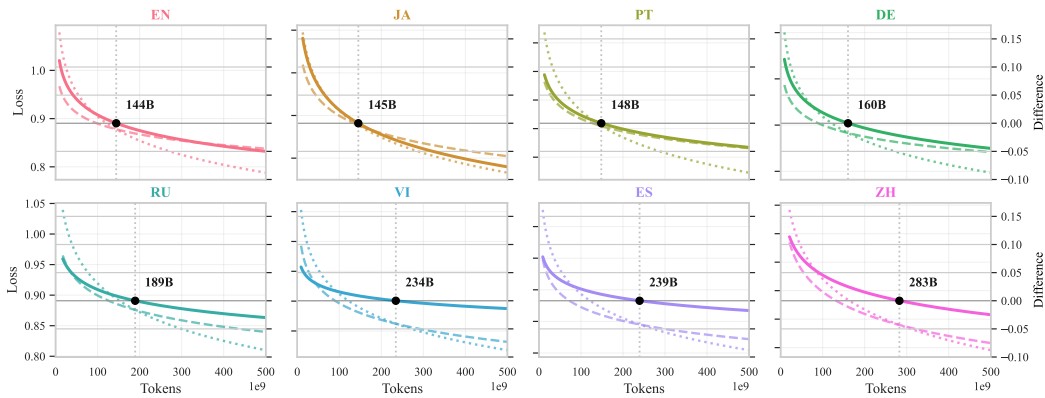

Figure 6: For eight languages, we plot the loss curves for 2B parameter models pretraining monolingually from scratch (···), finetuning monolingually from the Unimax base model (- - -), and the difference between these losses (—). We annotate the number of tokens at which the pretrained loss becomes better than the finetuning loss. The difference between the interpolated pretraining curve and finetuning curve. When the loss difference reaches zero, pretraining from scratch has surpassed finetuning using equal compute. **Depending on the token (or compute) budget, it is usually more effective to finetune a multilingual checkpoint if there are $< 144B$ tokens, and pretrain from scratch if the budget accommodates $\geq 283B$ tokens.**

## 6 PRETRAIN OR FINETUNE?

**Research Question:** *When training a model for target language $t$, is it more effective to pretrain from scratch, or finetune a general-purpose massively multilingual checkpoint?*

Prior scaling law work focuses on compute efficiency with respect to pretraining from scratch. However, in reality, practitioners who aim to optimize for a target language $t$ may have the option to start from multilingual public checkpoints or pretrain one multilingual checkpoint to serve as the starting point for many downstream models. Consider a practitioner with a compute budget $C$. In Figure 6, we plot the interpolated loss curves for the model pretrained from scratch, the model finetuned from a checkpoint, and the difference between these losses, for several languages. We find that while the warm-started finetuned model performs better at first, pretraining from scratch eventually surpasses its performance after $144B - 283B$ tokens, depending on the language. Note that we leave the reason for these convergence differences to future work, without a clear hypothesis. Though English pretraining converges the fastest, it is also allocated a 5% sampling rate within Unimax, whereas each of the other languages are allocated 1.4%.

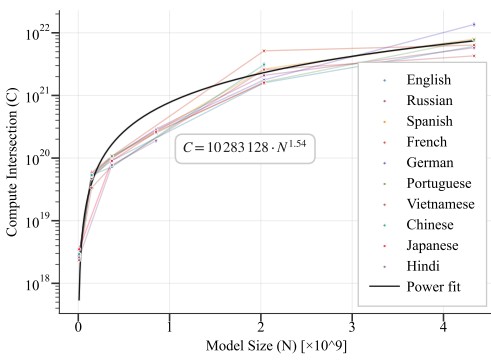

From the number of training tokens $D$ we can also infer the inflection point which decides whether it is more computationally efficient ($C = 6ND$) to pretrain from scratch or finetune from the multilingual checkpoint. Using our range of model size experiments, we can estimate a best fit power law in Figure 7, relating $N$ to $C$. We find $log(C) = 1113708 \times N^{1.65}$ to hold consistently across languages. For simplicity, we choose not to account for data repetition in this model. Practitioners may use this to estimate whether pretraining or finetuning will yield greater performance for their compute budget. Note one limitation to consider: not all multilingual base models are trained with the same mixture or for sufficiently long—and these factors would impact the pretrain vs finetuning intersection points. We choose a widely used Unimax mixture trained for 1B tokens (Chung et al., 2023). We believe this is a

Figure 7: In terms of $N$, we model the compute budget $C$ required for a model pretrained from scratch to outperform the model finetuned from the generic Unimax multilingual checkpoint. **We estimate this relationship as** $C = 10283128 \times N^{1.65}$.

useful heuristic, with reasonable assumptions, to determine the optimal choice between pretraining and finetuning for a given compute budget.

## 7 CONCLUSION

In this work, we answer several scaling questions on training multilingual models, validating our findings on scales up to 8B model parameters. First, we introduce a novel scaling law formulation called Adaptive Transfer Scaling Law (ATLAS) that significantly improves scaling law fit by more than 0.3 $R^2$ for multilingual mixtures. To aid our analysis, we also derive transfer scores for 38 langauages. We find that English is often a strong positive transfer language, alongside languages with shared script and language families, though we note that this transfer is often asymmetric. However, there is still a compute-efficiency tax when we train multilingual models. We quantify this "curse of multilinguality" and provide a mathematical trade-off between model capacity and language coverage: we find that the performance penalty from adding languages is mitigated more effectively by scaling model size ($N$) rather than training tokens ($D$). Finally, we quantitatively answer a critical practical question for practitioners: determining the compute crossover point between pretraining and finetuning. We find that depending on the language, for a budget under 144-283B tokens, finetuning a general-purpose multilingual checkpoint is more efficient. Collectively, these contributions provide valuable guidelines for practitioners determining data mixtures, language coverage and computational requirements for creating multilingual foundation models.

## 8 ACKNOWLEDGEMENTS

We thank Luke Zettlemoyer, Catherine Arnett and Stella Biderman for helpful discussions on the paper. We thank Biao Zhang and Xavier Garcia for the technical discussions and feedback on early directions.

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

# Appendix

## Table of Contents

# A EXTENDED RELATED WORK

**Scaling Laws** Scaling laws are used to study the behavior of deep learning systems on scaling training data and/or compute. Various researchers have observed scaling properties of generalization error with training data size and model capacity (Banko & Brill, 2001; Hestness et al., 2017; Amodei et al., 2016). Specifically, in the context of LLMs, Kaplan et al. (2020) explicitly propose a power law for the relationship between the loss $L$, number of parameters in the language model $N$, and the number of dataset tokens $D$. Later, Hoffmann et al. (2022) proposed a different formula:

$$L(N, D) = E + \frac{A}{N^\alpha} + \frac{B}{D^\beta} \tag{4}$$

We build upon this form throughout the paper to fit our scaling laws. Other researchers, too, have built upon these scaling laws to study various aspects of scaling, such as neural network architectures (Clark et al., 2022; Tay et al., 2022; Frantar et al., 2023; Gu & Dao, 2023; Scao et al., 2022) or transfer learning (Henighan et al., 2020). Researchers have also investigated scaling properties in different domains of deep learning such as neural machine translation (Ghorbani et al., 2021; Gordon et al., 2021), vision language models like CLIP (Cherti et al., 2023; Henighan et al., 2020), vision (Alabdulmohsin et al., 2022; Zhai et al., 2022), reinforcement learning (Hilton et al., 2023; Jones, 2021; Gao et al., 2023), and recommendation systems (Ardalani et al., 2022).

**Scaling Laws for Data Mixtures** Many prior works (Hoffmann et al., 2022) assume a fixed dataset distribution to study the effect of scaling on model performance. However, empirically, dataset composition can have a significant impact on quality (Longpre et al., 2023; Albalak et al., 2024; Sorscher et al., 2022). Hashimoto (2021) studies the relationship between dataset composition and model loss, finding a simple scaling law quantifying this relationship. Bansal et al. (2022) study the impact of data quality and noise on architecture, finding that synthetic data has a different exponent. Fernandes et al. (2023) study scaling for multilingual neural translation (Bapna et al., 2022), but experiments are restricted to 2-3 language pairs. Aghajanyan et al. (2023), in a similar vein, propose bimodal scaling laws for multimodal foundation models (Reid et al., 2024). Other researchers have considered different aspects of dataset composition that can affect model scaling: Muennighoff et al. (2024) study the effect of data repetition on scaling, while Goyal et al. (2024) consider how the effect of mixing data pools of varying quality changes with scale.

Foundation models involve training models on a mixture of datasets with the aim of achieving a certain validation loss on various validation sets and downstream datasets of interest. Based on the observation that optimal data mixture changes with scale, several recent works have attempted to characterize the change in the scaling law equation when either the train or test distribution changes.

Some researchers predict the optimal data mixture by first ablating mixtures using smaller models, and second, training a larger scaled-up model using the found optimal mixture (Ye et al., 2024; Xie et al., 2024; Liu et al., 2024). However, it is unclear if this optimal data mixture can extrapolate to a much larger magnitude, with some evidence in computer vision that this is not the case (Goyal et al., 2024) - i.e., the optimal data mixture changes with scale.

Some researchers have described scaling laws for different downstream evaluation sets for the same model (Dubey et al., 2024; Chen et al., 2024), while others (Brandfonbrener et al., 2024) have derived power law relationships between models trained on different distributions and the same model on different test sets.

Recently, several researchers have attempted to predict the final loss of a model for a given data mixture with scaling laws. Ge et al. (2024b) use the observation of a logarithmic shift in scaling pattern to predict loss on subsets of the Pile weighted in various proportions on a fixed model size. Ye et al. (2024), too, fit a scaling law for data mixtures based on a single scale. Kang et al. (2024) empirically show that the optimal mixture for a model changes with scale, and propose a method that addresses this on GPT-2 scale models. Que et al. (2024) use continual pre-training as a tool to develop their scaling law. Shukor et al. (2025), too, develop scaling laws that account for the transfer between data mixtures changing at scale for English LLMs and multimodal models for less than 10 domains.

**Multilingual Scaling Laws**   Prior work has also investigated multilingual transfer effects specifically in pretraining (Chang et al., 2024) and post-training (Shimabucoro et al., 2025), though on a smaller scale of models. He et al. (2024) is the work closest to ours, where the authors attempt to describe scaling laws for multilingual LLMs. A crucial distinction from our work is that we account for individual-language transfer learning in our scaling laws, and achieve a better resulting fit (Table 1).

## B   EXPERIMENTAL DETAILS

To understand multilingual scaling characteristics we run 774 separate experiments, across different model scales, and language mixtures. In Table B.1 we summarize the experiment segments, using the notation below, that defines sets of languages or scales.

- $\mathcal{L}_{\text{mono}}$ (7): {en, fr, ru, zh, hi, sw, yo}
- $\mathcal{L}_{\text{pairs}}$ (10): {en, fr, ru, zh, hi, de, es, pt, ja, vi}
- $\mathcal{L}_{\text{target}}$ (15): en/$X$ for $X \in \mathcal{L}_{\text{pairs}}$ + {es/pt, fr/zh, hi/ru, de/ja, hi/vi}
- $\mathcal{L}_{\text{eval}}$ (48): complete evaluation set (all 48 languages)
- $\mathcal{C}_{\text{exp}}$ (12): {$4 \times 4$, 6, $8 \times 2$, 12, 16, 24, 32, 50}
- $\mathcal{S}_{\text{full}}$ (20): {0, 1, 2, 4, 6, 8, 12, 14, 16, 18, 20, 22, 24, 26, 28, 30, 32, 34, 42, 46}
- $\mathcal{S}_{\text{part}}$ (11): {0, 4, 8, 12, 16, 20, 24, 28, 34, 42, 46}
- $\mathcal{S}_{\text{min}}$ (7): {0, 8, 16, 24, 34, 42, 46}

To see the model sizes defined by the scales in $\mathcal{S}_{\text{full}}$, $\mathcal{S}_{\text{part}}$, and $\mathcal{S}_{\text{min}}$, see Table B.3. We select these scale ranges to empirically observe models from $10M$ to $8B$ parameters. Subsets of languages, or subsets of the full scale ($\mathcal{S}_{\text{part}}$, or $\mathcal{S}_{\text{min}}$) are chosen to ensure the experiment number is sufficiently tractable for each research question, but also computationally feasible given our resources.

Table B.1: **An overview of experiment configurations in this work**. We enumerate the experiment types: **<Lang>** for monolingual scaling, **Unimax** as a massively multilingual baseline, **Language Pairs** to measure language-to-language transfer, **Capacity** to measure the curse of multilinguality, or model capacity constraints on learning new languages, and **Finetunes** to understand how finetuning from a massively multilingual model compares to pretraining from scratch. We use a mix of Monolingual and Multilingual vocabularies, and training data. In the LANGUAGES and SCALES columns we use parentheses to show the number of language mixtures and number of scales run. Symbols such as $\mathcal{L}_{\text{mono}}$ (7) and $\mathcal{S}_{\text{full}}$ (20) are defined above.

| EXPERIMENT TAG | VOCAB. | TRAIN-DATA | LANGUAGES | SCALES | # EXPS |
|---|---|---|---|---|---|
| **Mono <Lang>** | Monolingual | Monolingual | $\mathcal{L}_{\text{mono}}$ (7) | $\mathcal{S}_{\text{full}}$ (20) | $7 \times 20 = 140$ |
| **Unimax** | Multilingual | Multilingual | $\mathcal{L}_{\text{unimax}}$ | $\mathcal{S}_{\text{full}}$ (20) | 20 |
| **Multi <Lang>** | Multilingual | Monolingual | $\mathcal{L}_{\text{mono}}$ (7) | $\mathcal{S}_{\text{part}}$ (11) | $10 \times 7 = 70$ |
| **Language Pairs** | Multilingual | Monolingual | $\mathcal{L}_{\text{eval}}$ (48) | 34 | 50 |
| **Language Pairs** | Multilingual | Bilingual | $\mathcal{L}_{\text{pairs}}$ (10) | 34 | 90 |
| **Language Pairs** | Multilingual | Bilingual | $\mathcal{L}_{\text{target}}$ (15) | $\mathcal{S}_{\text{part}}$ (11) | $10 \times 15 = 150$ |
| **Capacity** | Multilingual | Uniform Sampling | $\mathcal{C}_{\text{exp}}$ (12) | $\mathcal{S}_{\text{part}}$ (11) | $10 \times 12 = 120$ |
| **Finetunes** | Multilingual | Monolingual | $\mathcal{L}_{\text{eval}}$ (48) | 34 | 50 |
| **Finetunes** | Multilingual | Monolingual | $\mathcal{L}_{\text{mono}}$ (7)$\cup\mathcal{L}_{\text{pairs}}$ (10) | $\mathcal{S}_{\text{min}}$ (7) | $7 \times 12 = 84$ |
| | | | | **Total** | 774 |

### B.1   LANGUAGE CHOICE

The fifty languages we selected come from the intersection of languages covered by both Flores-200 and MADLAD-400. They were selected to cover a wide range of language families, scripts, and resource levels. To get this collection, we sorted all languages in the Flores-MADLAD intersection by the number of characters, and then selected every six languages. We then went over this list and

perturbed the index up or down if it was too typologically similar to an already-selected language. For example, Language 48 was Afrikaans, but there were already several Germanic languages on the list, so we opted to take language 47 instead (Telugu).

## B.2 Experiment Design

**Pretraining Hyperparameter Details**   We rely on the hyperparameter settings refined by prior work to obtain reliable performance across model sizes. We also conducted initial experiments, varying batch sizes, learning rates, optimizer, and dropout to ensure our settings were relatively robust. In table B.2 we enumerate and explain all our hyperparameter choices, citing related work.

**Train & Test Data**   We use two test sets. For each we measure the vocabulary-insensitive loss proposed in Tao et al. (2024) in order to fairly evaluate across languages.

1. **MADLAD-400 Test** We isolate a test set from each of our 50 evaluation languages in MADLAD-400 (Kudugunta et al., 2024). In initial experiments we find stable metrics over random seeds requires sampling roughly (sequence length x num instances) $2048 * 10000 = 20,480,000$ tokens. For each language we randomly sample the minimum of either 20M tokens or 20% of the total tokens for low resource languages min(20M tokens, 20%).
2. **Flores-101** We use Flores-101 (Goyal et al., 2021), a language-parallel set of 101 high-quality translations, as a comparable test set. We extract the monolingual sequences the test set, using those to compute sequence losses, comparable to MADLAD-400 Test.

**Vocabulary Details**   We use a vocabulary trained by Kudugunta et al., with a 64000 sized vocabulary, $T = 100$ and 99.9995% character coverage.

## B.3 Experimental Details: Scaling Laws Evaluation

Typically, to evaluate fitted scaling laws, prior work has adopted the $R^2$ metric, for it's scale-invariant measure of a laws' explanatory power over the observed data. However, the choice of what data to hold-out as the test set can have a significant impact on the performance and it's interpretation (Li et al., 2025a). For this reason, and because multilingual scaling laws are used to measure more than just the predictive power of the law to larger scales, we use a set of different $R^2$ metrics to measure different objectives. These are:

1. $R^2$ Hold-out a randomly selected 20% of the data points from the data. This gives an impression of the models' ability to fit the variance, without measuring particular dimensions of fit.
2. $R^2(D)$ Hold-out the training runs with training tokens with the top 20% of $D$ tokens.
3. $R^2(N)$ — Hold-out a couple of scales of model sizes, including the largest: $N = 660M$ and $N = 8B$ parameters.
4. $R^2(C)$ Hold-out the largest compute parts of the training runs, where $C = 6ND$ FLOPs. This will include mainly a combination of the larger model sizes, and longer training runs.
5. $R^2(M)$ Hold-out all mixtures that are not monolingual, bilingual, or unimax. This allows the scaling law to get a baseline sense of each interacting language, but it has to infer the rest of the mixture scaling properties.

These metric variants allow us to unpack specifically where the scaling laws perform well, and for what purposes they are reliable.

## B.4 Experimental Details: Language Finetuning

In Section 6 we experiment with continuous pretraining on a single language, starting from a massively multilingual model's checkpoint. For clarity, we refer to this as *finetuning*, to distinguish it from pretraining from scratch.

We begin by training massively multilingual models, using UNIMAX (Chung et al., 2023) language sampling across all 420 languages in MADLAD-400. Chung et al. (2023) demonstrate this is an

Table B.2: **The pretraining hyperparameters used across experiments.** We detail the batch size scheduling, the vocabulary choice, as well as fine-grained hyperparameter choices. Each choice is justified and grounded in prior work, discussed in the Explanation column.

| PARAMETER | VALUE | EXPLANATION |
|---|---|---|
| *General Training Parameters* | | |
| Learning Rate Scheduler | WSD | We adopt the Warmup Stable Decay (WSD) learning rate schedule, designed to efficiently study data-model scaling law without extensive retraining (Hu et al., 2024; Hägele et al., 2024) |
| Optimizer | AdamW | Commonly used in scaling law experiments (Loshchilov & Hutter, 2019; Hoffmann et al., 2022) |
| Base Learning Rate | 2e-4 | Following (Muennighoff et al., 2024) and confirmation it works well in initial experiments. |
| Warmup Steps | 1000 | Standard practice, as in (Longpre et al., 2023) |
| Decay Period | 10% | Following (Hu et al., 2024; Hägele et al., 2024) this is the minimum well performing decay length. |
| Sequence Length | 2048 | As in (Longpre et al., 2023; Muennighoff et al., 2024). |
| Training Steps | 30k+ | We vary the steps according to experiments, to always ensure we are well above likely chinchilla compute optimal ranges. |
| Dropout | 0.1 | While English scaling laws work tends to use a dropout of 0.0, we see little notable difference, except for lower resource languages, with repeating epochs. |
| *Batch Size Schedule* | | |
| Batch Size (<150M params) | 256 | Hu et al. (2024); Muennighoff et al. (2024) both find the |
| Batch Size (<1B params) | 512 | optimal batch size increases with model size. We adopt a |
| Batch Size (<2B params) | 1024 | slightly larger rising batch size than Muennighoff et al. (2024), |
| Batch Size (2B+ params) | 2048 | following initial empirical results. |
| *Vocabulary* | | |
| Vocab Size | 64k | |
| Vocab (Unimax) | | Vocabulary by Kudugunta et al., with $T = 100$ and 99.9995% character coverage. |
| Vocab (Monolingual) | | Same settings as above, but trained only on sentences from the language being considered. |

effective sampling strategy to perform well across languages, as is the objective with many large multilingual models. The UNIMAX sampling rate per language is documented in Table B.4 Each of these models we train for $1T$ tokens, to reflect real-world practices.

Using these Unimax checkpoints, across model scales, we then conduct continuous monolingual pretraining (which we call *finetuning* here) on all 48 languages separately in $\mathcal{L}_{\text{eval}}$ (48). While finetuning, we also observe the loss across all $\mathcal{L}_{\text{eval}}$ (48)languages as well. We use the same hyperparameters as discussed for pretraining, but reset the learning schedule, and train for at least another $30k+$ steps. These empirical results allow us to compare (across a range of $N, D, C$) the loss on language $t$ between a model finetuned from a Unimax checkpoint, and a model pretrained monolingually from scratch on language $t$.

### B.5 EXPERIMENTAL DETAILS: LANGUAGE TRANSFER

In Section 4 we measure how much training on *source language s* is beneficial for the test loss on the *target language t*. A language transfer score tells a practitioner the extent to which training with each language would help or hinder their target language $t$. There are two ways in which we can measure this language transfer:

1. BILINGUAL TRANSFER SCORE. This score measures how many training tokens a 50/50 $(s, t)$ bilingual model needs, relative to a monolingual target language $t$ baseline, to reach the same validation loss $L_t$. A positive score indicates co-training with the source language has positive transfer, whereas a negative score indicates it hinders convergence.

Table B.3: **The SCALE value mapped to the dimensions of the model, and the parameter count.** The model dimensions, borrowed from Muennighoff et al. (2024), report the number of attention heads, number of layers, embed size, feedforward size, and key-value size. Our model may be slightly different sizes than prior work based on our vocabulary size (64000 + 512 special tokens).

| SCALE | HEADS | LAYERS | EMBED | FFW | KV | PARAMETERS |
|---|---|---|---|---|---|---|
| 0 | 4 | 3 | 128 | 512 | 32 | 9,044,352 |
| 1 | 7 | 4 | 224 | 896 | 32 | 17,662,848 |
| 2 | 7 | 5 | 288 | 1,152 | 32 | 24,847,776 |
| 3 | 7 | 6 | 448 | 1,792 | 32 | 45,763,200 |
| 4 | 8 | 8 | 512 | 2,048 | 64 | 66,588,672 |
| 5 | 9 | 9 | 576 | 2,304 | 64 | 84,939,840 |
| 6 | 10 | 10 | 640 | 2,560 | 64 | 106,830,080 |
| 7 | 10 | 13 | 640 | 2,560 | 64 | 126,492,800 |
| 8 | 10 | 16 | 640 | 2,560 | 64 | 146,155,520 |
| 9 | 12 | 12 | 768 | 3,072 | 64 | 162,800,640 |
| 10 | 12 | 15 | 768 | 3,072 | 64 | 191,114,496 |
| 11 | 12 | 18 | 768 | 3,072 | 64 | 219,428,352 |
| 12 | 14 | 14 | 896 | 3,584 | 64 | 237,646,080 |
| 13 | 14 | 16 | 896 | 3,584 | 64 | 263,337,984 |
| 14 | 14 | 18 | 896 | 3,584 | 64 | 289,029,888 |
| 15 | 16 | 16 | 1,024 | 4,096 | 64 | 334,512,128 |
| 16 | 16 | 18 | 1,024 | 4,096 | 64 | 368,068,608 |
| 17 | 16 | 20 | 1,024 | 4,096 | 64 | 401,625,088 |
| 18 | 10 | 18 | 1,280 | 5,120 | 128 | 554,457,600 |
| 19 | 10 | 21 | 1,280 | 5,120 | 128 | 633,104,640 |
| 20 | 11 | 18 | 1,408 | 5,632 | 128 | 661,807,872 |
| 21 | 10 | 24 | 1,280 | 5,120 | 128 | 711,751,680 |
| 22 | 11 | 21 | 1,408 | 5,632 | 128 | 756,970,368 |
| 23 | 12 | 19 | 1,536 | 6,144 | 128 | 816,345,600 |
| 24 | 11 | 24 | 1,408 | 5,632 | 128 | 852,132,864 |
| 25 | 12 | 22 | 1,536 | 6,144 | 128 | 929,596,416 |
| 26 | 12 | 25 | 1,536 | 6,144 | 128 | 1,042,847,232 |
| 27 | 14 | 20 | 1,792 | 7,168 | 128 | 1,143,245,824 |
| 28 | 14 | 23 | 1,792 | 7,168 | 128 | 1,297,391,872 |
| 29 | 14 | 26 | 1,792 | 7,168 | 128 | 1,451,537,920 |
| 30 | 16 | 22 | 2,048 | 8,192 | 128 | 1,608,560,640 |
| 31 | 17 | 22 | 2,176 | 8,704 | 128 | 1,807,137,536 |
| 32 | 16 | 25 | 2,048 | 8,192 | 128 | 1,809,893,376 |
| 33 | 16 | 28 | 2,048 | 8,192 | 128 | 2,011,226,112 |
| 34 | 17 | 25 | 2,176 | 8,704 | 128 | 2,034,422,912 |
| 35 | 18 | 24 | 2,304 | 9,216 | 128 | 2,187,122,688 |
| 36 | 17 | 28 | 2,176 | 8,704 | 128 | 2,261,708,288 |
| 37 | 18 | 28 | 2,304 | 9,216 | 128 | 2,526,870,528 |
| 38 | 18 | 32 | 2,304 | 9,216 | 128 | 2,866,618,368 |
| 39 | 20 | 26 | 2,560 | 10,240 | 128 | 2,891,514,880 |
| 40 | 20 | 30 | 2,560 | 10,240 | 128 | 3,310,955,520 |
| 41 | 20 | 34 | 2,560 | 10,240 | 128 | 3,730,396,160 |
| 42 | 21 | 36 | 2,688 | 10,752 | 128 | 4,335,303,168 |
| 43 | 22 | 36 | 2,816 | 11,264 | 128 | 4,749,364,224 |
| 44 | 23 | 36 | 2,944 | 11,776 | 128 | 5,182,299,648 |
| 45 | 24 | 36 | 3,072 | 12,288 | 128 | 5,634,109,440 |
| 46 | 28 | 40 | 3,584 | 14,336 | 128 | 8,452,190,208 |
| 47 | 32 | 42 | 4,096 | 16,384 | 128 | 11,538,702,336 |
| 48 | 32 | 47 | 4,352 | 17,408 | 128 | 14,314,315,520 |
| 49 | 36 | 44 | 4,608 | 18,432 | 128 | 15,245,973,504 |
| 50 | 32 | 47 | 4,608 | 18,432 | 128 | 15,821,655,552 |
| 51 | 32 | 47 | 4,864 | 19,456 | 128 | 17,402,920,192 |
| 52 | 40 | 47 | 5,120 | 20,480 | 128 | 18,982,943,616 |

Table B.4: **The Unimax language sampling rates adapted from Chung et al. (2023).** Languages are listed in order of their percentage sampling rate, which sum to 100.

| LANG | % | LANG | % | LANG | % | LANG | % | LANG | % | LANG | % |
|---|---|---|---|---|---|---|---|---|---|---|---|
| en | 5.00e+00 | af | 1.42e+00 | be | 1.42e+00 | fil | 1.42e+00 | gl | 1.42e+00 | te | 1.42e+00 |
| de | 1.42e+00 | es | 1.42e+00 | fr | 1.42e+00 | hr | 1.42e+00 | is | 1.42e+00 | kk | 1.42e+00 |
| ml | 1.42e+00 | mr | 1.42e+00 | ru | 1.42e+00 | sr | 1.42e+00 | ta | 1.42e+00 | az | 1.42e+00 |
| hi | 1.42e+00 | id | 1.42e+00 | it | 1.42e+00 | lv | 1.42e+00 | ms | 1.42e+00 | nl | 1.42e+00 |
| pl | 1.42e+00 | pt | 1.42e+00 | sq | 1.42e+00 | sv | 1.42e+00 | tr | 1.42e+00 | vi | 1.42e+00 |
| ca | 1.42e+00 | et | 1.42e+00 | hu | 1.42e+00 | iw | 1.42e+00 | ro | 1.42e+00 | sl | 1.42e+00 |
| th | 1.42e+00 | zh | 1.42e+00 | ar | 1.42e+00 | cs | 1.42e+00 | fa | 1.42e+00 | fi | 1.42e+00 |
| ja | 1.42e+00 | ko | 1.42e+00 | lt | 1.42e+00 | sk | 1.42e+00 | uk | 1.42e+00 | bg | 1.42e+00 |
| da | 1.42e+00 | el | 1.42e+00 | no | 1.42e+00 | mk | 1.38e+00 | bn | 1.34e+00 | eu | 1.34e+00 |
| ka | 1.19e+00 | mn | 1.09e+00 | bs | 1.03e+00 | uz | 1.02e+00 | ur | 8.19e-01 | sw | 7.19e-01 |
| ne | 6.87e-01 | kaa | 6.76e-01 | kn | 6.72e-01 | gu | 6.43e-01 | si | 5.89e-01 | cy | 5.14e-01 |
| eo | 5.06e-01 | la | 4.64e-01 | hy | 4.47e-01 | ky | 4.37e-01 | tg | 4.28e-01 | ga | 4.25e-01 |
| mt | 4.06e-01 | my | 3.95e-01 | km | 3.35e-01 | tt | 3.14e-01 | so | 2.93e-01 | ps | 2.52e-01 |
| ku | 2.50e-01 | pa | 2.38e-01 | rw | 2.29e-01 | lo | 2.06e-01 | dv | 1.83e-01 | ha | 1.78e-01 |
| ckb | 1.73e-01 | fy | 1.68e-01 | lb | 1.63e-01 | mg | 1.54e-01 | ug | 1.52e-01 | am | 1.50e-01 |
| gd | 1.48e-01 | ht | 1.27e-01 | grc | 1.25e-01 | jv | 1.12e-01 | tk | 1.09e-01 | hmn | 1.09e-01 |
| sd | 1.05e-01 | mi | 9.77e-02 | yi | 9.55e-02 | ba | 9.42e-02 | fo | 9.24e-02 | ceb | 9.12e-02 |
| or | 9.07e-02 | kl | 8.12e-02 | xh | 7.21e-02 | su | 7.20e-02 | ny | 6.97e-02 | sm | 6.94e-02 |
| sn | 6.68e-02 | co | 6.67e-02 | pap | 6.57e-02 | zu | 6.46e-02 | ig | 6.31e-02 | yo | 6.00e-02 |
| st | 5.70e-02 | haw | 5.38e-02 | as | 5.07e-02 | oc | 4.93e-02 | cv | 4.66e-02 | lus | 4.61e-02 |
| tet | 4.14e-02 | gsw | 4.04e-02 | sah | 4.01e-02 | br | 3.29e-02 | rm | 2.52e-02 | sa | 2.23e-02 |
| bo | 2.23e-02 | om | 2.22e-02 | se | 2.12e-02 | ce | 1.70e-02 | cnh | 1.58e-02 | ilo | 1.49e-02 |
| hil | 1.44e-02 | udm | 1.39e-02 | os | 1.26e-02 | lg | 1.21e-02 | ti | 1.12e-02 | vec | 1.10e-02 |
| ts | 9.73e-03 | tyv | 9.66e-03 | kbd | 9.23e-03 | ee | 8.25e-03 | iba | 7.66e-03 | av | 7.57e-03 |
| kha | 7.57e-03 | to | 7.51e-03 | tn | 7.33e-03 | nso | 7.08e-03 | fj | 7.02e-03 | zza | 6.60e-03 |
| ak | 6.23e-03 | ada | 6.08e-03 | otq | 5.86e-03 | dz | 5.69e-03 | bua | 5.44e-03 | cfm | 5.41e-03 |
| ln | 5.39e-03 | chm | 5.36e-03 | gn | 5.23e-03 | krc | 5.21e-03 | wa | 5.11e-03 | hif | 4.79e-03 |
| yua | 4.32e-03 | srn | 4.26e-03 | war | 4.03e-03 | rom | 3.99e-03 | bik | 3.94e-03 | sg | 3.89e-03 |
| lu | 3.87e-03 | ady | 3.73e-03 | kbp | 3.68e-03 | syr | 3.51e-03 | ltg | 3.49e-03 | myv | 3.48e-03 |
| iso | 3.43e-03 | kac | 3.43e-03 | bho | 3.38e-03 | ay | 3.30e-03 | kum | 3.10e-03 | qu | 3.06e-03 |
| pag | 3.02e-03 | ngu | 2.97e-03 | ve | 2.94e-03 | pck | 2.88e-03 | zap | 2.86e-03 | tyz | 2.83e-03 |
| hui | 2.73e-03 | bbc | 2.65e-03 | tzo | 2.65e-03 | tiv | 2.55e-03 | ksd | 2.52e-03 | gom | 2.50e-03 |
| min | 2.47e-03 | ang | 2.46e-03 | nhe | 2.45e-03 | bgp | 2.45e-03 | nzi | 2.37e-03 | nnb | 2.29e-03 |
| nv | 2.28e-03 | bci | 2.26e-03 | kv | 2.25e-03 | new | 2.21e-03 | mps | 2.19e-03 | alt | 2.18e-03 |
| meu | 2.15e-03 | bew | 2.13e-03 | fon | 2.08e-03 | iu | 2.08e-03 | abt | 2.07e-03 | mgh | 2.05e-03 |
| tvl | 2.02e-03 | dov | 2.00e-03 | tlh | 1.96e-03 | ho | 1.96e-03 | kw | 1.92e-03 | mrj | 1.92e-03 |
| meo | 1.89e-03 | crh | 1.89e-03 | mbt | 1.87e-03 | emp | 1.85e-03 | ace | 1.85e-03 | ium | 1.85e-03 |
| mam | 1.81e-03 | gym | 1.74e-03 | mai | 1.72e-03 | crs | 1.70e-03 | pon | 1.69e-03 | ubu | 1.68e-03 |
| quc | 1.62e-03 | gv | 1.57e-03 | kj | 1.49e-03 | btx | 1.48e-03 | ape | 1.46e-03 | chk | 1.45e-03 |
| rcf | 1.44e-03 | shn | 1.42e-03 | tzh | 1.41e-03 | mdf | 1.39e-03 | ppk | 1.38e-03 | ss | 1.37e-03 |
| gag | 1.31e-03 | cab | 1.27e-03 | kri | 1.25e-03 | seh | 1.23e-03 | ibb | 1.23e-03 | tbz | 1.21e-03 |
| bru | 1.21e-03 | enq | 1.20e-03 | ach | 1.17e-03 | cuk | 1.16e-03 | kmb | 1.15e-03 | wo | 1.14e-03 |
| kek | 1.12e-03 | qub | 1.11e-03 | tab | 1.11e-03 | bts | 1.07e-03 | kos | 1.06e-03 | rwo | 1.05e-03 |
| cak | 1.05e-03 | tuc | 1.02e-03 | bum | 1.01e-03 | gil | 9.73e-04 | stq | 9.65e-04 | tsg | 9.47e-04 |
| quh | 9.39e-04 | mak | 9.37e-04 | arn | 9.35e-04 | ban | 9.06e-04 | jiv | 8.84e-04 | sja | 8.49e-04 |
| yap | 8.33e-04 | tcy | 8.26e-04 | toj | 8.20e-04 | twu | 8.15e-04 | xal | 8.08e-04 | amu | 8.05e-04 |
| rmc | 8.04e-04 | hus | 7.83e-04 | nia | 7.77e-04 | kjh | 7.73e-04 | bm | 7.64e-04 | guh | 7.64e-04 |
| mas | 7.64e-04 | acf | 7.56e-04 | dtp | 7.46e-04 | ksw | 7.25e-04 | bzj | 7.22e-04 | din | 7.17e-04 |
| zne | 7.15e-04 | mad | 6.99e-04 | msi | 6.84e-04 | mag | 6.60e-04 | mkn | 6.57e-04 | kg | 6.54e-04 |
| lhu | 6.37e-04 | ch | 6.23e-04 | qvi | 5.78e-04 | mh | 5.59e-04 | djk | 5.52e-04 | sus | 5.21e-04 |
| mfe | 5.20e-04 | srm | 5.12e-04 | dyu | 5.08e-04 | ctu | 5.07e-04 | gui | 5.03e-04 | pau | 5.02e-04 |
| inb | 4.88e-04 | bi | 4.71e-04 | mni | 4.59e-04 | guc | 4.43e-04 | jam | 4.39e-04 | wal | 4.38e-04 |
| jac | 4.35e-04 | bas | 4.30e-04 | gor | 4.18e-04 | skr | 4.15e-04 | nyu | 4.14e-04 | noa | 4.08e-04 |
| sda | 4.08e-04 | gub | 4.07e-04 | nog | 4.04e-04 | teo | 4.01e-04 | tdx | 3.92e-04 | sxn | 3.81e-04 |
| rki | 3.76e-04 | nr | 3.74e-04 | frp | 3.61e-04 | alz | 3.60e-04 | taj | 3.51e-04 | lrc | 3.50e-04 |
| cce | 3.22e-04 | rn | 3.21e-04 | jvn | 3.14e-04 | hvn | 3.08e-04 | nij | 3.07e-04 | dwr | 2.99e-04 |
| izz | 2.79e-04 | msm | 2.78e-04 | bus | 2.73e-04 | ktu | 2.66e-04 | chr | 2.52e-04 | maz | 2.39e-04 |
| tzj | 2.22e-04 | suz | 2.15e-04 | knj | 2.15e-04 | bim | 1.99e-04 | gvl | 1.98e-04 | bqc | 1.98e-04 |
| tca | 1.97e-04 | pis | 1.92e-04 | laj | 1.83e-04 | qxr | 1.82e-04 | niq | 1.80e-04 | ahk | 1.80e-04 |
| shp | 1.78e-04 | hne | 1.75e-04 | spp | 1.71e-04 | koi | 1.67e-04 | quf | 1.53e-04 | agr | 1.39e-04 |
| tsc | 1.31e-04 | mqy | 1.27e-04 | gof | 1.27e-04 | gbm | 1.25e-04 | miq | 1.22e-04 | dje | 1.21e-04 |
| awa | 1.19e-04 | qvz | 1.10e-04 | tll | 1.03e-04 | raj | 1.02e-04 | kjg | 1.00e-04 | quy | 9.24e-05 |
| cbk | 8.52e-05 | akb | 8.48e-05 | oj | 8.46e-05 | ify | 8.39e-05 | cac | 8.03e-05 | brx | 7.64e-05 |
| qup | 7.48e-05 | ff | 6.97e-05 | ber | 6.68e-05 | tks | 6.55e-05 | trp | 6.46e-05 | mrw | 6.46e-05 |
| adh | 6.40e-05 | smt | 6.16e-05 | ffm | 5.93e-05 | qvc | 5.87e-05 | ann | 5.40e-05 | kaa_Latn | 5.26e-05 |
| nut | 4.62e-05 | kwi | 4.34e-05 | msb | 4.20e-05 | el_Latn | 4.09e-05 | doi | 3.40e-05 | dln | 2.97e-05 |
| hi_Latn | 2.48e-05 | ctd_Latn | 1.03e-05 | ru_Latn | 6.95e-06 | te_Latn | 4.84e-06 | ber_Latn | 4.74e-06 | az_RU | 3.23e-06 |
| ta_Latn | 2.29e-06 | tly_IR | 2.15e-06 | nan_Latn_TW | 1.94e-06 | ml_Latn | 1.80e-06 | zxx_xx_dtynoise | 1.65e-06 | gom_Latn | 1.28e-06 |
| bg_Latn | 9.21e-07 | kn_Latn | 6.18e-07 | zh_Latn | 5.69e-07 | cr_Latn | 2.59e-07 | bn_Latn | 1.83e-07 | gu_Latn | 1.69e-07 |
| sat_Latn | 1.51e-07 | ndc_ZW | 1.31e-07 | kmz_Latn | 1.01e-07 | ms_Arab | 6.00e-08 | ms_Arab_BN | 4.29e-08 | | |

2. FINETUNING ADAPTATION SCORE. This score captures the effect on the target language's loss $L_t$, when a massively multilingual model is finetuned only on the source language $s$.

### B.5.1 BILINGUAL TRANSFER SCORE (BTS)

The Bilingual Transfer Score asks how data-efficient a bilingual learner is, relative to a purely monolingual one, on some target language $t$. To estimate this, we pretrain two models, a monolingual one on $t$ and a bilingual one that samples tokens equally (50/50) from the source and target languages $(s, t)$. We then measure how many more training tokens it takes the bilingual model to attain the same validation loss $L_t$ as the monolingual model. This "distance" between the learning curves is measured throughout training, at different reference horizons—or, number of tokens trained on. In Figure 2 we use models with 2B parameters, and a reference horizon of $d = 42B$ tokens, as it roughly equates to $10,000$ pretraining steps for our 2B parameter models, and learning curves have stabilized. In Figure C.1 we measure how the BTS varies by model size, and by number of training tokens seen.

We are able to compute the bilingual transfer scores, across every combination of pairings between 10 languages: English, Russian, Spanish, French, German, Portuguese, Chinese, Vietnamese, Japanese, and Hindi. These languages were chosen to give a distribution of high- and mid-resource languages from a variety of language families. This totals 10 monolingual experiments (one for each language), and $C(10, 2) = 45$ bilingual experiments, which provide the results for a 10x10 grid of Bilingual Transfer Scores.

In more detail, let $d_{mono}$ be the number of tokens trained on language $t$ by a monolingual model. This model attains test loss of $L_t(d_{mono})$ after $d_{mono}$ tokens on language $t$. We record the number of training tokens $d_{bi}$ at which the bilingual model reaches the same loss, $L_t(d_{bi}) = L_t(d_{mono})$. We define the **Bilingual Transfer Score** (BTS) as the signed step surplus

$$\text{BTS}_{s \to t} = -\frac{\sigma_{\text{bi}}(L_{\text{t}}(d_{\text{mono}})) - 2d_{\text{mono}}}{d_{\text{mono}}}$$

where $d_{\text{bi}} = \sigma_{\text{bi}}(L_{\text{t}}(d_{\text{mono}}))$. $\sigma_{\text{bi}}$ maps a loss value to the number of steps taken by the bilingual model to reach that loss. In short, this score measures the relative training efficiency of the bilingual model compared to a monolingual model at reaching the same loss level for language $t$. Subtracting $2d_{\text{mono}}$ and dividing by $d_{\text{mono}}$ simply centers the value on 0, and forces positive transfer to give a positive value. A value of 0, implies *neutral* transfer, as the multilingual model requires exactly the same number of steps in the target language $t$ as the monolingual model, a value below 0 implies *negative* transfer, as the multilingual model is slower to reach the loss $L_{\text{t}}(d_{\text{mono}})$, and a value above 0 implies *positive* transfer, as the bilingual model reaches the target loss in $< 2d_{\text{mono}}$. Note that for Figure 2 these scores are anchored on 2B parameter models, trained for $d = 42B$ tokens, for consistency. But as model size increases, or tokens trained decreases, the language transfer scores improve monotonically.

### B.5.2 FINETUNING ADAPTATION SCORE (FAS)

Complementary to BTS, the Finetuning Adaptation Score measures the *instantaneous impact* of continued exposure to a single language on all others while holding model capacity fixed. Specifically, it measures how finetuning a massively multilingual model on source language $s$ affects the performance of target language $t$. Because we only need to train on each source language once, and evaluate on all languages, it is less computationally expensive than BTS, which requires training on every pair of languages. As such, we are able to derive $38 * 38 = 1444$ $(s, t)$ comparisons from 38 finetuning experiments.

We start from a shared multilingual checkpoint: a Unimax pretrained model, trained for $1T$ tokens. For each source language $s$, we continue pre-training (for simplicity we denote this as *finetune*) exclusively on this language, and evaluate at regular intervals on every target language's $t \in \mathcal{L}$ validation set. Let $L_{s \to t}(d)$ be the validation loss on $t$ after $d$ additional updates on $s$, and let $L_t^{unimax}$ denote the baseline loss of the shared Unimax checkpoint, before finetuning has begun. The Finetun-

ing Adaptation Score (FAS) aggregates the reduction in loss over a common time window $[0, d_{\max}]$:

$$\text{FAS}_{s \to t} = \frac{1}{d_{\max}} \int_0^{d_{\max}} \left[ L_t^{unimax} - L_{s \to t}(d) \right] \mathrm{d}d.$$

Positive values indicate that exposing the model to $s$ accelerates learning or yields immediate gains on $t$ ("helpful transfer"), whereas negative values capture negative interference. Normalising by $d_{\max}$ makes the score comparable across language pairs and training horizons.

Because finetuning on language $s$ can cause unpredictable behavior in the validation loss of a target language $t$ (it might immediately increase, or decrease then eventually increase), we start with deriving a series of features about the finetuning loss curve. We calculate the following features for a finetuning loss curve:

- **Baseline Loss Deviation ($\delta_{s \to t}$).** For each language pair $s \neq t$ we quantify the deviation in validation loss between the baseline $L_t^{unimax}$ and the loss $L_{s \to t}$ on the final step:

$$\delta_{s \to t} = L_{s \to t}(d_{\max}) - L_t^{unimax}(d_{\max})$$

  $\delta_{s \to t} < 0$ means fine–tuning on $s$ helps $t$; $\delta_{s \to t} > 0$ means it hurts.

- **Adaptation Gain ($g_l$).** This measures the area normalized reduction in loss on language $l$ from finetuning on language $l$. This allows us to capture the relative learning dynamics of both the source and target language.

$$g_l = \frac{1}{d_{max}} \int_0^{d_{\max}} \left[ L_l^{unimax} - L_{s \to t}(d) \right] \mathrm{d}s.$$

  $g_l > 0$ indicates net improvement; large values imply that $l$ benefits greatly from additional supervision.

For each language transfer pair $(s, t)$, we compose the following feature vector $x_{s \to t}$. As languages differ in intrinsic difficulty, we normalize each feature.

$$\tilde{g}_l = \frac{g_l - \mu_g}{\sigma_g}, \qquad\qquad \tilde{\delta}_{s \to t} = \frac{\delta_{s \to t} - \mu_{\delta,t}}{\sigma_{\delta,t}}, \qquad\qquad (5)$$

For adaptation gain, we use the global mean and standard deviation $(\mu_g, \sigma_g)$, and for baseline loss deviation we compute the mean and standard deviation $(\mu_{\delta,t}, \sigma_{\delta,t})$ across all source languages $s$ for each fixed target language $t$.

## B.6 Estimating Bilingual Transfer Score (BTS)

In the Bilingual Transfer Score experiments we empirically measured language transfer scores $y_{s \to t}^{\text{true}}$ for a subset $\mathcal{P}_{\text{obs}} \subset \mathcal{L} \times \mathcal{L}$. We construct a training set from these 90 experiments, using them as the gold labels.

$$\mathbf{x}_{s \to t} = \left[ \tilde{g}_s, \ \tilde{g}_t, \ \tilde{g}_s - \tilde{g}_t, \ \tilde{\delta}_{s \to t} \right]^\top, \qquad y_{s \to t} = y_{s \to t}^{\text{true}}, \qquad (6)$$

We fit a random-forest regressor $\varphi_{\boldsymbol{\theta}} : \mathbb{R}^4 \to \mathbb{R}$ with 300 trees and unlimited depth. The predictive performance is assessed by $k$-fold cross-validation ($k = 5$), obtaining an $R^2 = 0.85$ and Spearman correlation of $\rho = 0.88$. After training on all observed pairs, $\varphi$ is invoked on every unlabelled $(s, t) \in \mathcal{L} \times \mathcal{L}$, yielding predicted BTS values, that estimate the language transfer shown in Figure 2.

## B.7 Experimental Details: Multilingual Capacity

### B.7.1 Multilingual Capacity Scaling Laws

In Section 5 we aim to understand how a model's capacity hinders multilingual learning—also termed the *curse of multilinguality* (Conneau et al., 2019; Chang et al., 2024). To do this, we train models of various sizes, and with a range of language mixtures, shown in Table B.5. For simplicity, languages are always evenly sampled within their mixtures, even if some languages have more available text than others. We train models on these mixtures, and evaluate the validation loss of all other languages, throughout training.

Table B.5: **The set of experiments designed to measure the *curse of multilinguality*—or how language performance is impacted by both the number of training languages and the model size.** We defined VERSION mixtures with NUM languages, ranging between 4 and 50. For mixtures with 4 languages, we define a few different versions oriented to languages that might be more likely to be grouped together. This set of experiments are summarized in the **Capacity** row of Table B.1.

| VERSION | NUM | LANGUAGES (ALPHABETICAL) |
|---|---|---|
| **4v0** | 4 | de, es, fr, pt |
| **4v1** | 4 | de, en, es, fr |
| **4v2** | 4 | hi, ja, vi, zh |
| **4v3** | 4 | en, hi, pt, ru |
| **6v0** | 6 | de, en, es, fr, pt, ru |
| **8v0** | 8 | de, en, es, fr, pt, ru, vi, zh |
| **8v1** | 8 | de, en, fr, hi, ja, ru, vi, zh |
| **12v0** | 12 | de, en, es, fr, hi, it, ja, pl, pt, ru, vi, zh |
| **16v0** | 16 | de, en, es, fr, hi, id, it, ja, nl, pl, pt, ru, sv, tr, vi, zh |
| **24v0** | 24 | cs, da, de, en, es, fa, fi, fr, hi, hu, id, it, ja, nl, pl, pt, ro, ru, sv, th, tr, uk, vi, zh |
| **32v0** | 32 | ar, bg, ca, cs, da, de, el, en, es, fa, fi, fr, hi, hu, id, it, ja, ko, lt, nl, no, pl, pt, ro, ru, sk, sv, th, tr, uk, vi, zh |
| **50v0** | 50 | af, ar, ba, bm, bn, ca, ckb, cy, de, ee, el, en, es, fa, fil, fr, gn, gu, ha, he, hi, hr, id, it, ja, jv, kk, ko, mr, ms, my, nl, no, om, pl, ps, pt, ro, ru, sm, sv, sw, te, th, tr, uk, ur, vi, yo, zh |

For a given target language $t$ we have empirical loss scores across 11 differently sized models (from 10M to 8B), and across each mixture that contains the language in training, including monolingual, bilingual, and the multilingual mixtures shown in Table B.5.

We model each language's loss as $L(N, D, K)$, where $N$ is the model size, $D$ are the number of training tokens for the *target* language, and $K$ is the number of languages in the training mixture (evenly sampled). We define $L(N, D, K)$ as follows:

$$L(N, D, K) \;=\; L_\infty \;+\; \frac{A\,K^\phi}{N^\alpha} \;+\; \frac{B\,K^\psi}{D^\beta} \tag{7}$$

This law preserves the well-validated power-law behavior of monolingual scaling laws, while introducing $K$ into both the capacity term ($AK^\phi/N^\alpha$) and the data term ($BK^\psi/D^\beta$). The exponent $\phi$ captures how capacity requirements grow with the number of languages, while $\psi$ captures how data requirements change with multilinguality: $\psi < 0$ indicates *positive transfer* (sublinear per-language data needs as $K$ increases), $\psi = 0$ implies no net transfer in the data term, and $\psi > 0$ implies negative transfer/interference. Under even sampling, the total tokens across languages are $D_{\text{tot}} = K \cdot D$, in which case the data term can be rewritten as $B\,K^{\psi+\beta}/D_{\text{tot}}^{\beta}$.

### B.7.2 ADDING LANGUAGES: $K$ TO $K * r$

Practitioners may wish to change the number of languages supported by a model, from $K$ to $K * r$. If the model size and training tokens remain unchanged, it will lead to a degradation in loss across languages, as the same compute budget is allocated across more languages. When adding languages, the practitioner may wish to know how much they need to scale the model ($N$, $D$) and the training compute $C$ to maintain the same performance per language, as before. We call this an *iso-loss*, as it approximately preserves the loss values, with a change in $K$.

**Compute-Optimality for Equation (7).** As we increase $K$ to $K * r$ we would like to understand the new $N'$, $D'$ that *minimize* training compute on the iso-loss surface. Let $r = K'/K$. Minimizing $C' \propto N'D'$ subject to

$$\frac{A(Kr)^\phi}{N'^\alpha} + \frac{B(Kr)^\psi}{D'^\beta} \;=\; \frac{AK^\phi}{N^\alpha} + \frac{BK^\psi}{D^\beta}$$

yields the closed-form optimum:

$$\left(\tfrac{N'}{N}\right)^\star = r^{\phi/\alpha}, \qquad \left(\tfrac{D'}{D}\right)^\star = r^{\psi/\beta}, \qquad \left(\tfrac{D'_{tot}}{D_{tot}}\right)^\star = r^{1+\psi/\beta}.$$

These expressions show a clear separation of roles: $\phi/\alpha$ governs the parameter burden of adding languages, whereas $\psi/\beta$ governs the data burden (with $\psi < 0$ reducing the per-language data requirement due to positive transfer). Since the compute budget is in terms of total tokens $D_{\text{tot}} = K \cdot D$, and $C \propto N D_{\text{tot}}$.

**Adding languages: $K$ to $K * r$ — Finding the compute multiplier.** At the iso-loss optimum, the compute multiplier is

$$\frac{C'}{C} = \frac{N' D'_{tot}}{N D_{tot}} = \left(\frac{N'}{N}\right)^{\star} \left(\frac{D'_{\text{tot}}}{D_{\text{tot}}}\right)^{\star} = r^{1+\phi/\alpha+\psi/\beta}.$$

Intuitively, $\frac{\phi}{\alpha}$ measures the extra parameter capacity per added language, while $\frac{\psi}{\beta}$ measures the extra (or reduced, if $\psi < 0$) per-language token budget. But since compute is defined with total tokens $D_{\text{tot}}$, then we consider the additional factor of $r$ from enlarging the language inventory.

**Adding languages: $K$ to $K * r$ — Finding the Iso-loss curve.** Let $s \equiv \frac{N'}{N}$ and $t \equiv \frac{D'}{D}$ denote the multipliers for model size and (per-target-language) data when we grow the language inventory from $K$ to $K' = rK$. Define the baseline term contributions

$$w_N \equiv \frac{AK^{\phi}/N^{\alpha}}{AK^{\phi}/N^{\alpha} + BK^{\psi}/D^{\beta}}, \qquad w_D = 1 - w_N.$$

The iso-loss condition equates the *sum of the two error terms* before and after the change in $K$:

$$\frac{A(Kr)^{\phi}}{(Ns)^{\alpha}} + \frac{B(Kr)^{\psi}}{(Dt)^{\beta}} = \frac{AK^{\phi}}{N^{\alpha}} + \frac{BK^{\psi}}{D^{\beta}}.$$

Dividing both sides by $AK^{\phi}/N^{\alpha} + BK^{\psi}/D^{\beta}$ yields the normalized *iso-loss constraint*

$$r^{\phi} w_N s^{-\alpha} + r^{\psi} w_D t^{-\beta} = 1. \tag{8}$$

**Solving for $t$ given $s$.** Rearranging equation 8 gives

$$t = \left[\frac{r^{\psi} w_D}{1 - r^{\phi} w_N s^{-\alpha}}\right]^{1/\beta}.$$

The denominator must be positive, which implies the feasibility condition

$$s^{\alpha} > r^{\phi} w_N.$$

As $s^{\alpha} \downarrow r^{\phi} w_N$ the denominator tends to $0^+$ and $t \to \infty$; as $s \to \infty$, the denominator tends to 1 and $t \to (r^{\psi} w_D)^{1/\beta}$.

**Solving for $s$ given $t$.** By symmetry,

$$s = \left[\frac{r^{\phi} w_N}{1 - r^{\psi} w_D t^{-\beta}}\right]^{1/\alpha}, \qquad \text{with feasibility } t^{\beta} > r^{\psi} w_D.$$

**Total-token form.** Under even sampling, $D_{\text{tot}} = KD$ and $D'_{\text{tot}} = K'D' = rKD'$, so

$$\frac{D'_{\text{tot}}}{D_{\text{tot}}} = r \frac{D'}{D} = r \left[\frac{r^{\psi} w_D}{1 - r^{\phi} w_N s^{-\alpha}}\right]^{1/\beta}.$$

**Compute-optimal frontier (starting from a compute-optimal $(K, N, D)$).** Suppose the *baseline* $(K, N, D)$ is compute-optimal for its loss level, i.e., it minimizes $C \propto ND$ subject to $AK^{\phi}/N^{\alpha} + BK^{\psi}/D^{\beta} = \text{const}$. A standard Lagrange multiplier argument then yields

$$\alpha \frac{AK^{\phi}}{N^{\alpha}} = \beta \frac{BK^{\psi}}{D^{\beta}} \iff w_N = \frac{\beta}{\alpha + \beta}, \quad w_D = \frac{\alpha}{\alpha + \beta}.$$

After changing $K \to rK$, the compute along the iso-loss curve is $C'/C = s\, t$ (or $C'/C = s \cdot \frac{D'_{\text{tot}}}{D_{\text{tot}}}$ if compute is defined with total tokens). Minimizing this product subject to equation 8 again via Lagrange multipliers gives the stationarity condition

$$\alpha\, r^{\phi} w_N s^{-\alpha} = \beta\, r^{\psi} w_D t^{-\beta}.$$

Table C.1: **A summary of the monolingual scaling laws for each language** The columns are: the language, the language family, the language script, its number of unique tokens $|D|$ in MADLAD-400 (with the percentage of unique tokens as compared to English), and the derived scaling law.

| LANGUAGE | FAMILY | SCRIPT | $|D|$ (% EN) | SCALING LAW |
|---|---|---|---|---|
| *Monolingual Vocabulary, Monolingual Training* | | | | |
| English | Indo-european | Latin | 2.8T (100.0%) | $0.67 + \frac{e^{6.18}}{N^{0.41}} + \frac{e^{8.25}}{D^{0.41}}$ |
| French | Indo-european | Latin | 363B (13.01%) | $0.76 + \frac{e^{10.11}}{N^{0.64}} + \frac{e^{13.24}}{D^{0.64}}$ |
| Russian | Indo-european | Cyrillic | 705B (25.26%) | $0.82 + \frac{e^{10.07}}{N^{0.63}} + \frac{e^{13.28}}{D^{0.63}}$ |
| Chinese | Sino-tibetan | Hans | 125B (4.48%) | $1.08 + \frac{e^{8.20}}{N^{0.46}} + \frac{e^{10.37}}{D^{0.46}}$ |
| Hindi | Indo-european | Devanagari | 7.9B (0.28%) | $0.52 + \frac{e^{6.03}}{N^{0.39}} + \frac{e^{8.03}}{D^{0.39}}$ |
| Swahili | Niger-congo: atlantic congo | Latin | 770M (0.03%) | $0.00 + \frac{e^{4.13}}{N^{0.26}} + \frac{e^{5.51}}{D^{0.26}}$ |
| *Multilingual Vocabulary, Monolingual Training* | | | | |
| English | Indo-european | Latin | 2.8T (100.0%) | $0.83 + \frac{e^{9.91}}{N^{0.63}} + \frac{e^{13.06}}{D^{0.63}}$ |
| French | Indo-european | Latin | 363B (13.01%) | $0.66 + \frac{e^{7.12}}{N^{0.45}} + \frac{e^{8.94}}{D^{0.45}}$ |
| Russian | Indo-european | Cyrillic | 705B (25.26%) | $0.76 + \frac{e^{7.97}}{N^{0.49}} + \frac{e^{9.91}}{D^{0.49}}$ |
| Chinese | Sino-tibetan | Hans | 125B (4.48%) | $1.18 + \frac{e^{8.87}}{N^{0.49}} + \frac{e^{10.90}}{D^{0.49}}$ |
| Hindi | Indo-european | Devanagari | 7.9B (0.28%) | $0.63 + \frac{e^{6.99}}{N^{0.43}} + \frac{e^{8.69}}{D^{0.43}}$ |
| Swahili | Niger-congo: atlantic congo | Latin | 770M (0.03%) | $0.00 + \frac{e^{4.99}}{N^{0.30}} + \frac{e^{6.43}}{D^{0.30}}$ |
| *Multilingual Vocabulary, Unimax Training* | | | | |
| English | Indo-european | Latin | 2.8T (100.0%) | $0.00 + \frac{e^{3.03}}{N^{0.16}} + \frac{e^{3.15}}{D^{0.16}}$ |
| French | Indo-european | Latin | 363B (13.01%) | $0.00 + \frac{e^{3.63}}{N^{0.19}} + \frac{e^{3.94}}{D^{0.19}}$ |
| Russian | Indo-european | Cyrillic | 705B (25.26%) | $0.00 + \frac{e^{3.90}}{N^{0.20}} + \frac{e^{4.28}}{D^{0.20}}$ |
| Chinese | Sino-tibetan | Hans | 125B (4.48%) | $0.39 + \frac{e^{4.58}}{N^{0.21}} + \frac{e^{5.47}}{D^{0.21}}$ |
| Hindi | Indo-european | Devanagari | 7.9B (0.28%) | $0.00 + \frac{e^{3.46}}{N^{0.18}} + \frac{e^{3.89}}{D^{0.18}}$ |
| Swahili | Niger-congo: atlantic congo | Latin | 770M (0.03%) | $0.00 + \frac{e^{3.52}}{N^{0.18}} + \frac{e^{3.74}}{D^{0.18}}$ |

Combining with equation 8 and substituting the compute-optimal weights above yields the *unique* minimum-compute point on the new frontier:

$$\left(\frac{N'}{N}\right)^{\star} = r^{\phi/\alpha}, \qquad \left(\frac{D'}{D}\right)^{\star} = r^{\psi/\beta}, \qquad \left(\frac{D'_{\text{tot}}}{D_{\text{tot}}}\right)^{\star} = r^{1+\psi/\beta}.$$

At this point each error component is *individually restored* to its baseline magnitude: $A(Kr)^{\phi}/N'^{\alpha} = AK^{\phi}/N^{\alpha}$ and $B(Kr)^{\psi}/D'^{\beta} = BK^{\psi}/D^{\beta}$, so the weights $(w_N, w_D)$—and hence the balance of capacity/data bottlenecks—remain unchanged. Because the frontier is convex in $(\log s, \log t)$ and $\log C' = \log s + \log t$ is linear, this stationary point is the global compute minimum along the iso-loss curve.

## C  EXTENDED RESULTS

### C.1  EXTENDED RESULTS: SCALING LAWS

In Table C.1 we illustrate the fitted scaling laws for each language. Specifically, we show the scaling law parameters for each language when trained with a monolingual vocabulary on monolingual data, when trained with a multilingual vocabulary on monolingual data, and when trained with a multilingual vocabulary on the Unimax multilingual mixture. These results match up with those presented in Figure 1.

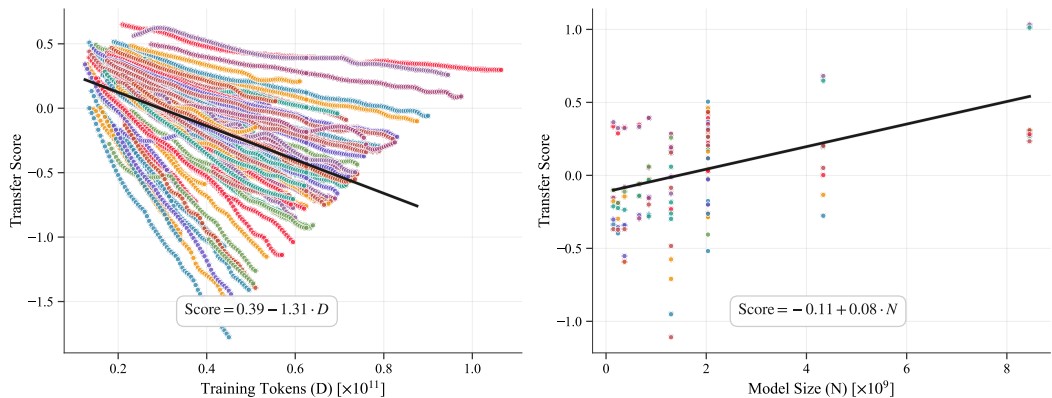

Figure C.1: **Left:** X-axis is training tokens (D), Y-axis is language-transfer score. **Right:** X-axis is model size (N), Y-axis is language-transfer score.

## C.2 EXTENDED RESULTS: LANGUAGE TRANSFER

**Research Question:** *How does language transfer change with larger models, or when you train for longer?*

We further investigated how language transfer evolves with training data (D) and model size (N) in Figure C.1. Our analysis of transfer scores over the course of training reveals that transfer dynamics are established early. The performance gap between synergistic pairs (e.g., es → pt) and interfering pairs (e.g., en → zh) appears within the initial stages of training and remains largely consistent, suggesting that the fundamental compatibility of languages is not significantly altered by longer training. However, the impact of model scale is more pronounced. As illustrated in Figure C.1, we observe a clear trend in which larger models are more effective in facilitating cross-lingual transfer. For synergistic pairs, the positive transfer score increases modestly with model capacity. More importantly, for challenging pairs that exhibit interference in smaller models, larger models are significantly better at mitigating this negative effect, bringing the score closer to zero. This is similar to what has been observed while training larger and larger foundation models (Team et al., 2024). Given that transfer dynamics change at scale, it is an important factor to consider when designing multilingual foundation models that are performant for all languages being trained on.

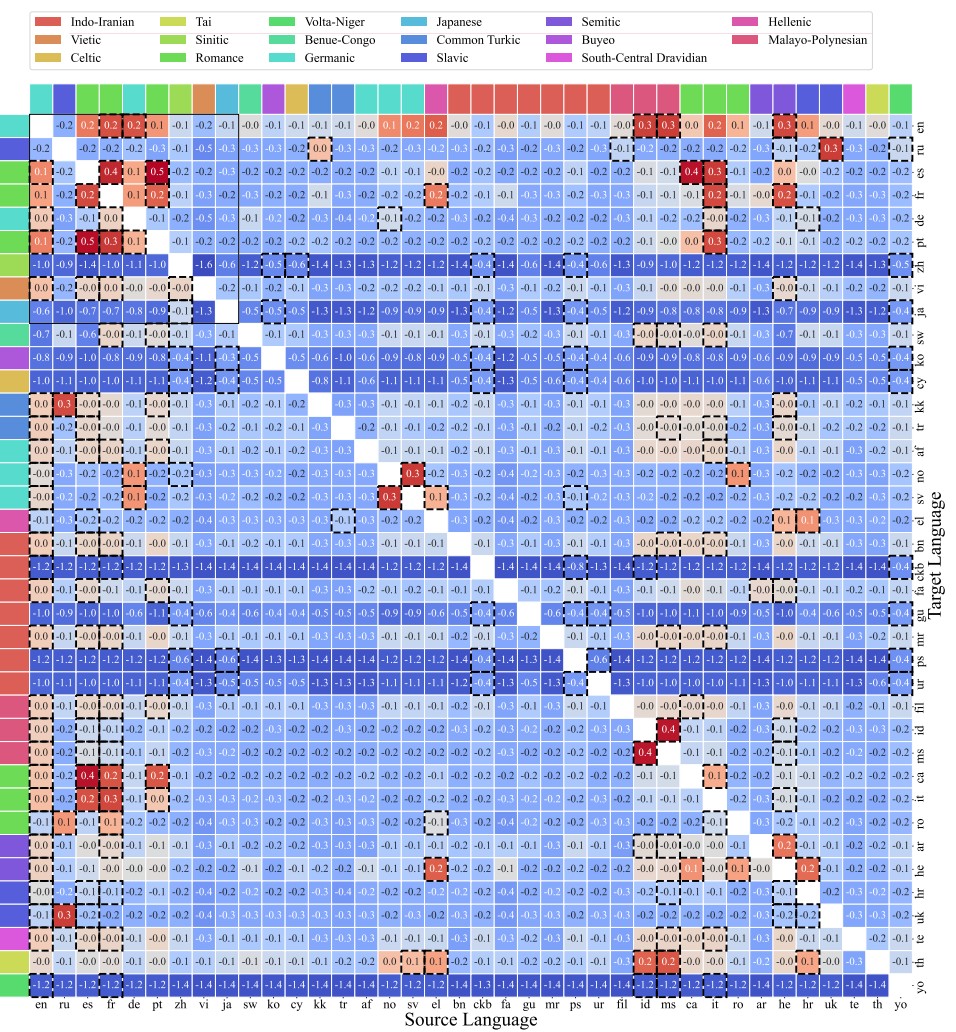

Figure C.2: The language transfer scores measure the benefit to the target language of (co-)training with the source language. We bold the top-5 source languages for each target language. The top left 9x9 are the Bilingual Transfer Score computed directly, whereas the rest are estimated from the Finetuning Adaptation Score. While English is the best source language for many of of languages, we find language similarity is highly predictive of these scores.

