# OpenReview forum: "ATLAS: Adaptive Transfer Scaling Laws for Multilingual Pretraining, Finetuning, and Decoding the Curse of Multilinguality"
_ICLR.cc/2026/Conference — ICLR 2026 Poster_

### Official Review · Reviewer_Yyfb · 2025-11-03

**Soundness:** 3
**Presentation:** 3
**Contribution:** 3
**Rating:** 6
**Confidence:** 4

**Summary:**

1. The paper presents scaling laws in the multilingual setup across different axis:

1.1 For repeated epochs in the monolingual setup

1.2 To account for cross lingual transfer in a language mixture setup

1.3. Account for the curse of multilinguality by providing a closed form approximation to account for how much more data to train / how many more parameters to account for to keep the loss on a target language consistent, while adding new languages.

1.4. Address the issue when is it better to pretrain a language model for a target language from scratch vs finetune a multilingual language model.

2. The authors demonstrate that their scaling laws fit better for monolingual and multilingual scenarios compared to other monolingual, data constrained and multilingual scaling laws respectively. In order to account for scaling across different languages with different vocabularies, they fit scaling laws to the vocabulary insensitive loss.

3. The authors also present a very large scale cross lingual transfer study. They propose a cross lingual transfer metric: the number of target language tokens taken by a bilingual model to reach the same loss as that taken by the target language monolingual model. By analysing the transfer matrix, the authors highlight the key factors associated with positive transfer: language script as well as language family. They also demonstrate that these factors additionaly seem predictive of whether a symmetric transfer might occur between the languages.

4. The authors present their results over an impressive number of experiments. If the results (esp. details on number of parameters, tokens, loss convergence values, mixture weights, data subset etc.) are released, it can enable additional analysis especially w.r.t cross-lingual transfer.

**Strengths:**

1. The paper presents a new functional form for modeling multilingual setups, accounting for repeated tokens as well as data mixtures. The consequent scaling law has better predictive power compared to other baselines.

2. For me, the biggest contribution is the cross-lingual study for understanding transfer at scale: the significant number of pairwise experiments definitely help identify key factors for cross-lingual transfer, and additionally be an important resource for trying to further understand what factors might actually help influence the degree of the transfer.

3. The paper's answers an important question on how should I scale compute (N,D) to achieve the same loss level while adding new languages. They also provide a good heuristic approximation on when it might be better to fine-tune a pre-trained multilingual language model vs train the lm from scratch.

4. The authors present their results over an impressive number of experiments: 750 independant training runs. Managing experiments at this scale can be and is difficult, and the effort is appreciated.

**Weaknesses:**

While I think the paper make some really good contributions, I do have the following concerns:

1. The scaling law's functional form doesn't explain why it was chosen to be the functional form in the first place, compared to other ways of formulating the interaction. Concretely [1,2], both demonstrate that L(N, D, p) ~ L(N, D)*p exp(\gamma), which also requires that the term modeling the parameters should also depend on the proportion of language (a similar symmetrization argument was also made in [3]).  Some motivation on why the functional form was chosen would be good to have: both for the general purpose law as well as the target language specific loss (Page 7: L373).

2. For the section on curse of multilinguality, the authors mention that they sample the tokens from each language uniformly: however that is counter to the sampling setup used on most other multilingual experiments (Unimax). For consistancy, it would be good to know if the laws also hold under the unimax distribution setup. Orthogonally, the authors give a recommendation on how to scale compute (N, D) for accommodating new languages. But without actual verification of the scaling law (i.e actually training with the additional compute on the new languages), it is hard to verify the accuracy of the proposed law. Concretely, it would be informative to have a setup to train with languages not used for the fitting, but belonging to either a group with positive transfer, neutral transfer or negative transfer, and then showing how accurate the law is at predicting the performance.

3. The section on pre-train vs fine-tune seems very rushed, and is a bit hard to follow. Additionally, it is quite counter-intuitive that the proposed inflection point is not a function of the compute capacity spent on the pretrained unimax model. Overall, as a meta comment, I think this section requires a bit more of a rigorous setup to compare (eg: varying C for the pre-trained model). In addition to that, there are a number of observations from the cross-lingual setup that are difficult to explain: eg: for the target language of En, why do source languages id and ms have a high BTS score. Having a better understanding of that might be very informative.


[1] He, Yifei, et al. "Scaling laws for multilingual language models." arXiv preprint arXiv:2410.12883 (2024).

[2] Akimoto, Kosuke, and Masafumi Oyamada. "Optimizing Low-Resource Language Model Training: Comprehensive Analysis of Multi-Epoch, Multi-Lingual, and Two-Stage Approaches." arXiv preprint arXiv:2410.12325 (2024).

[3] Muennighoff, Niklas, et al. "Scaling data-constrained language models." Advances in Neural Information Processing Systems 36 (2023): 50358-50376.

**Questions:**

1. For leveraging the vocabulary agnostic loss, how did that compare against fitting without the load agonstic setup. Does this design choice have any implication on the results presented in the paper? Likewise, were the baseline MSL fit on the vocabulary agnostic loss or on general log-likelihood?

2. Somewhat related to the previous question: what is your opinion on balancing out the vocabulary at a per language level [1], and leveraging that for loss computation (instead of doing a vocabulary agnostic loss) ?

3. For the MSL evaluations, what was the language grouping that was used for computing the scaling laws for fitting ? Also as a meta point, it would be good to also add details on how the curve fitting was actually carried out.

4. If my understanding of Eqn (3) is correct, the authors make the assumption that the languages decay at the same rate. Intuitively, the decay rate (\lambda) should be a function of the dataset quality, which varies significantly across languages. So why would the assumption hold true ?

5. For Figure 1, hi, seems like overtraining hurts both the monolingual and the unimax model, however the monolingual model with the multilingual vocabulary seems to be doing okay. Why do the authors think that might be the case?


[1] Zheng, Bo, et al. "Allocating large vocabulary capacity for cross-lingual language model pre-training." arXiv preprint arXiv:2109.07306 (2021).

---

> ### Author Response · Authors · 2025-11-25
>
> We thank Reviewer Yyfb for their thoughtful and encouraging feedback. We are pleased they highlighted the scope of our contributions—from the cross-lingual transfer matrix to scaling (N, D) with languages and optimizing finetuning vs. pretraining—and we appreciate their recognition of the improved predictive power of our scaling laws.
>
> **High-level response.** Your comments focus on the assumptions and generalization limits of our scaling laws. We agree and will reinforce the discussion with additional paragraphs to help practitioners understand the context in which the laws apply. As with all scaling-law work—especially when modeling complex relationships—certain assumptions are unavoidable, including (but not limited to) training and hyperparameter setup, model height/width scaling, learning schedule, *vocabulary*, training-data choice and filters, *data mixture laws*, *evaluation metric*, and the *heuristic relationship between (C, N, D)*. We have italicized the items you emphasized. Our goal is to make these assumptions explicit and to justify them as reasonable for the questions we study. While we do not have compute for additional experiments, we believe this work breaks ground on several complex questions. Below, we clarify our choices and propose targeted revisions to justify assumptions and limitations.
>
> **Why did we choose ATLAS’s functional form?**
>
> Yes, we can expound on this in the paper further! Beyond following the Chinchilla scaling-law form, we build on the effective data formulation introduced by Muennighoff et al. (2023) for single repeated mixtures, and then introduce a new formulation, similar to He et al. (2024), but that breaks down data contributions from the target and transfer languages in the mixture. While we don’t encode data proportions explicitly in ATLAS, we do encode the number of tokens (and repeated tokens) per target and transfer language, each weighted by their own $\lambda$. This should allow the law to capture varied mixture proportions.
>
> We experimented with quite a few variants (including with parameters also weighted by proportions) but wanted to err on the side of something simpler than more complex. We just re-ran a variant now of ours that also normalizes by $p*exp(gamma)$ for the overall data and model terms, as done in He et al. (2024), and results degraded slightly. That isn’t to say that there exists a stronger formulation, but we value ATLAS for its joint effectiveness and relative simplicity as a starting point.
>
> We would be happy to (a) motivate these choices more explicitly in the paper, and (b) add a couple other variants for comparison. (Let us know if you have one you are particularly interested in?)
>
> In Section 5, we chose a simple Chinchilla form, but added the K interaction with separate exponents for data and model sizes. This is again simple, preserves Chinchilla power-law behaviors, and ($\phi$, $\psi$) allow us to interpret relative scaling interactions between N,D,K. We will add this motivation, as we believe this law is very well motivated in preserving existing power-law behaviors, having a good fit, but also giving us interpretable results.
>
> In Section 6 we just use a simple power-law $C=A*N^\alpha$ since there is no $D$. We will clarify the distinction between these uses in the revised draft.
>
> [1] https://arxiv.org/abs/2305.16264
>
> **Section 5 (Curse of Multilinguality): Why uniform sampling rather than unimax? Does evaluation hold out new language mixtures?**
>
> We agree that unimax sampling is common in practice. Our study explicitly targets a different applied setting: a practitioner building a model to serve a specific subset of languages (e.g., 1, 2, 4, 8, or 12) and then scaling within that subset. In such cases, uniform sampling across the chosen languages is a plausible operational assumption. We will edit the text to state this assumption and application more clearly.
>
> Regarding R² evaluations, we do hold out training runs corresponding to particular data mixtures (e.g., a four-language mixture) when computing fit quality. This procedure aligns with the validation setting you describe.
>
> **Section 6 (FT vs. PT): Should we model the trade-off in terms of pretraining compute?**
>
> This would indeed improve the analysis, but we propose leaving it for future work (we are out of compute budget). Our claim: the pretraining budget we chose should be robust and reasonable to inform many training settings. We fixed the pretraining budget in Section 6 to 1T tokens. Across the model sizes we study, these checkpoints are over-trained under Chinchilla, providing strong fits; practitioners are unlikely to prefer substantially under-trained checkpoints, and further over-training would yield diminishing returns. Given your feedback, we will justify this choice more rigorously and explicitly acknowledge other variables that can affect the scaling law (including this one and those listed in the High-level response).

---

> > ### Author Response · Authors · 2025-11-25
> >
> > **Vocabulary agnostic loss: How did that compare against fitting without the load agnostic setup. Does this design choice have any implication on the results presented in the paper?**
> >
> > Good question. We recorded non-vocabulary-agnostic losses as well in our experiments, so we just tested the results using the general log perplexity instead of vocabulary-agnostic (bot for MSL and our own). We actually don’t really see any significant changes in aggregate metrics, just small variations of a couple percent R^2 in either direction.
> >
> > For instance, the Chinchilla Scaling Laws in a multilingual setting change from this to this:
> >
> > Before: —-----------------------------------
> >
> > 0.64  -0.99  0.73  0.66  0.61
> >
> > After: —-----------------------------------
> >
> > 0.64  -0.76  0.69  0.67  0.62
> >
> > **MSL evals, what language grouping was used? Details on curve fitting?**
> >
> > We use the same language groupings as in the MSL paper, but extend them across more language families, as documented by Kudugunta et al (2024) in MADLAD-400.
> >
> > For curve-fitting we adapt the parametric fit functions from this work: https://github.com/huggingface/datablations/blob/main/utils/parametric_fit.ipynb. Specifically we use `torch.optim.LBFGS`, and we’re happy to release this fitting code in the camera ready.
> >
> > [1] https://arxiv.org/abs/2309.04662
> >
> > **Equation 3: we assume languages decay at the same rate (lambda): is this assumption actually true?**
> >
> > Thank you for pointing out this typo - we have updated the draft. We use different lambda for each language, which is more intuitive.
> >
> > **For Figure 1, why are repeating epochs not causing all of the Hindi optimal trajectories to curve upwards?**
> >
> > This is a great question, and honestly we believe it is likely noise in the data, and slightly less reliable fit for the largest sizes of Hindi’s multi-mono curve. There are fewer data points at the largest scale, so it may not have accounted for diminishing returns as well in that setting. From the upward tilt in all 3 Swahili curves, the 2 other Hindi curves, and even the last part of the 3 Chinese curves, we believe this is a clear phenomena, with a little noise. We will point this out as an anomaly in the data.
> >
> > **Finally, we’d like to thank you again for your detailed and constructive feedback. We believe and hope these changes address your core critiques. Please let us know what additional changes we can make to improve the standing of this work in your review. Thank you!**

---

### Official Review · Reviewer_taDw · 2025-11-10

**Soundness:** 1
**Presentation:** 1
**Contribution:** 1
**Rating:** 2
**Confidence:** 3

**Summary:**

This paper studies the scaling law of multi-lingual models w.r.t. model size, data size and computation budgets.

**Strengths:**

• Systematic and comprehensive study of scaling law for multilingual models is an important topic.
	• Significant number of experiments are conducted. A few important findings are drawn (the evidence to support the claims requires additional attention though).

**Weaknesses:**

• Key definitions are missing for a few key concepts and key equations. For example, the symbols in equation (1) are not defined. The grounding of these symbols and equations are not available in this paper. The readers will need to look up the citations with significant efforts of guessing to understand the key idea in an inaccurate manner.
	• The formal model of scaling laws are not well defined in this paper.
	• The writing and structure of this paper doesn't meet the scientific paper quality.

**Questions:**

1. Line 015, in what sense is this the largest multilingual scaling law study?
	2. Line 075, please define N, D, C before they used? I am guessing they are model size, data size, computational buduget? Especially the symbol N is difficult to bet against.
	3. Line 093-097, as R^2 is a key concept in this paper, it is worth explaining/re-iterating the intuition behind R^2 on model size, token numbers, computational costs and number of pairs of languages (M --- I am again guessing the meaning of this symbol).
	4. Line 104, will the vocab size affect the analysis results? Is 64K vocab size too small for 48 languages especially with glyphic scripted languages.
	5. Line 159 - Line 178, please expand the description to include formal definitions, intuitive explanation and other details which can help readers to understand the design of ATLAS. It would be more rigorous if this can be grounded on any statistical models that can draw relationship among the variables in ATLAS and draw relationships on these variables and statistical data collected from model training experiments regarding validity of the proposal.
	6. Line 197, how do you choose C=6ND?
	7. Line 1208, "well-validated power-law behavior", please provide references and evidence in terms of statistical data fitting for experical training tasks. As the LLM training behavior is highly complex, it needs substantial evicence to draw conclusion that simple linear or power-law equations can fit the training behaviors with stats over data size, model size, language number, and computation cost.
	8. Line 483-485, any conclusion or summary section?

---

> ### Author Response · Authors · 2025-11-24
>
> Dear Reviewer, we would like to thank you for your time and feedback. We would welcome constructive feedback on later sections of the paper and the overall scientific merit of the work.
>
> **Scaling Law Definitions**
>
> Note that we already do define N,D, and C in the Experimental Setup (now lines 95-105). And they, along with the Chinchilla formulation (Equation 1), are widely used in scaling laws research. Per your request, we have updated the paper to include more explicit definitions and interweave relevant citations. We have also added a conclusion section to highlight our contributions and recommendations for practitioners.
>
> Below we answer your questions:
>
> *“in what sense is this the largest multilingual scaling law study?”*
>
> Compared to other works on multilingual scaling (He et al, 2024 is the closest to ours), we conduct experiments on twice the number of languages and 7x the maximum model size. Moreover, compared to prior work on data mixtures, we also use significantly more data mixtures while testing our scaling laws - Please refer to Appendix A for an extended related work and comparison.
>
> [1] https://arxiv.org/abs/2410.12883
>
> *“will the vocab size affect the analysis results?”*
>
> 64k is a vocabulary size that has been found to be appropriate for multilingual models in literature (Bapna et al, 2022) for even more languages, even when including glyphic scripted languages. Hence, we have fixed 64k as our vocabulary size as a tradeoff between quality and computational cost. Given that the effect of vocabulary on loss has been found to be a modest overall scaling factor (Tao et al, 2024), we believe that provided that the chosen vocabulary size is not so small that the quality degrades, the overall analysis will remain the same.
>
> [1] https://arxiv.org/abs/2205.03983
>
> [2] https://arxiv.org/abs/2407.13623
>
> *“how do you choose C=6ND?”*
>
>  C=6ND is a well-known approximation for the computational cost in FLOPs for training models. We direct the review to Porian et al 2024, Li et al 2024 for a discussion of the same.
>
> [1] https://arxiv.org/pdf/2502.18969
>
> [2] https://arxiv.org/abs/2406.19146
>
> *”well-validated power-law behavior”*
>
> In our work, we empirically observe that all our data mixtures do follow a power law in accordance with related literature. In literature, other researchers too have found that scaling laws do hold over different data mixtures - we point the reviewer to Appendix A for more related work on data mixtures, and Villalobos, 2023 or  Li et al 2023 for a discussion of the various settings under which scaling laws hold.
>
> [1] https://epoch.ai/blog/scaling-laws-literature-review
>
> [2] https://arxiv.org/pdf/2502.18969
>
> We believe that these changes address your feedback, and would appreciate an appropriate increase in scores.

---

### Official Review · Reviewer_gXoZ · 2025-11-11

**Soundness:** 3
**Presentation:** 3
**Contribution:** 4
**Rating:** 8
**Confidence:** 4

**Summary:**

The paper proposes a new multilingual scaling framework that models how performance in multilingual language models scales with model size, data size, and the number of languages during pretraining and finetuning. To understand cross-lingual transfer, the work provides a large-scale empirical study quantifying the pairwise language transfer in a model-based manner. In addition, it models and quantifies the curse of multilinguality, providing scaling rules for maintaining performance as language coverage expands. Finally, the work analyzes when it is more efficient to pretrain from scratch compared with finetuning from a multilingual checkpoint.

**Strengths:**

- The problems the paper wants to tackle are important in the multilingual learning literature. Each section begins with a clear research question, which guides the reader through the narrative logically.

- The work shows significant experimental efforts. Notably, the bilingual transfer table in Figure 2 is a valuable asset to the community of multilingual learning.

- The findings offer actionable insights for multilingual model practitioners, especially the compute-optimal scaling frontier and pretrain-vs-finetune tradeoff. These results help translate empirical observations into practical guidance for scaling multilingual models efficiently.

**Weaknesses:**

- It is unclear which part of the proposed law is a unique contribution of the authors, and which is adapted. For instance, it is known that equation (1) essentially follows the Chinchilla scaling law, but why do equations (2) and (3) take the given specific form? Although the fitting accuracy is high, it would be helpful to provide some justification about why the proposed law organizes those terms in the optimal way. If those constructions are an improvement or combination of previous scaling laws, it would be better to first explicitly write their corresponding forms and name the improvement or generalization of the proposed law mathematically.

- The practical use of the proposed scaling law for predicting model performance or guiding training design remains somewhat implicit. In section 5 and 6, how is the proposed law used? A clearer description of the parameter fitting process would also strengthen the work: for example, what data points are used to estimate the coefficients, what optimization method is applied, and how stable the fitted parameters are across runs. Furthermore, the computational cost of fitting the scaling law is a key practical factor. If estimating these parameters requires comparable or greater resources than training a target model directly, the law’s real-world usefulness would be limited.

- The paper lacks a conclusion section that summarizes the findings or discusses future directions. The narrative ends abruptly after the analysis, making the work feel unfinished.

**Questions:**

- In section 5, the study of the curse of multilinguality, the authors do not seem to explicitly model the effect of inter-language interaction. For instance, if the added languages are highly similar to the existing ones, the curse of multilinguality might not be as evident. On the other hand, if the added language is very different, the effect would be more pronounced. Does the current analysis take this language similarity into account?

- The transfer between pairs of languages is clearly modeled in Figure 2. What about transfer in terms of subsets of languages? The effect of cross-lingual transfer for a subset may not merely manifest as the sum of transfer in each of the constituent languages as multiple languages combined could show redundancy or synergy. How would this be captured by the current scaling law?

- For finetuning discussed in section 6, do the authors actually mean continual pretraining? If so, it might be a better term as finetuning could be confused with SFT in the context of LLM, causing confusion.

---

> ### Author Response · Authors · 2025-11-24
>
> We would like to thank Reviewer gXoZ for their constructive feedback and their strong support of our submission! We appreciate that they recognize our significant experimental efforts, and the actionable insights for practitioners: especially the compute-optimal frontier and pretrain-vs-finetuning trade-offs. We agree with all your feedback, and propose specific changes to address them below.
>
> **Why did we choose ATLAS’s functional form?**
>
> Yes, we can expound on this in the paper further! Beyond following the Chinchilla scaling-law form, we build on the effective data formulation introduced by Muennighoff et al. (2023) for single repeated mixtures, and then introduce a new formulation for effective data in mixtures. We try to incorporate transfer effects in the most simple, tractable way, as compared to prior work (Appendix A). We will add there may be improved formulations from data mixture research we did not test, but this is a strong simple starting point.
>
> In Section 5, we chose a simple Chinchilla form, but added the K interaction with separate exponents for data and model sizes. This is again simple, preserves Chinchilla power-law behaviors, and ($\phi$, $\psi$) allow us to interpret relative scaling interactions between N,D,K.
>
> In Section 6 we just use Chinchilla, but then to estimate the fine-tuning vs pretraining inflection point, we use a simple power-law $C=A*N^\alpha$ since it is only in terms of model size $N$ (inflection point $D$ can be calculated later from $C$). We will clarify the distinction between these uses in the revised draft.
>
> **Section 5-6: In section 5 and 6, how is the proposed law used?**
>
> Sorry yes, we can clarify this more explicitly. In Sections 5 and 6 we do not re-use ATLAS, but use different simplified laws to model specific other phenomena: the number of languages K, and when monolingual pretraining surpasses monolingual finetuning.
>
> **A clearer description of the parameter fitting process would also strengthen the work**
>
> For curve-fitting we adapt the parametric fit functions from this work: https://github.com/huggingface/datablations/blob/main/utils/parametric_fit.ipynb. Specifically we use `torch.optim.LBFGS`, and we’re happy to release this fitting code in the camera ready.
>
>
> **Furthermore, the computational cost of fitting the scaling law is a key practical factor.**
>
> Generally, a full scaling law sweep can be obtained with 10-20% of the training budget of the final model. This is likely feasible for most practitioners. In fact, Porian et al show that a good scaling law estimate can be obtained with even less FLOPs, improving cost-effectiveness.
>
> However, we think our learned scaling laws can also provide general boundaries that inform training runs for MADLAD-400 and beyond, saving multilingual practitioners compute in guiding the bounds of their own experiments or final runs.
>
> [1] https://arxiv.org/abs/2406.19146
>
>
> **Section 5: Why don’t we explicitly model inter-language interaction here?**
>
> There are a couple assumptions here (lines 359, 363-365) that we will state more clearly from the outset: (1) we sample uniformly from training languages, and (2) we add languages roughly in descending order of resourcefulness. Given the massive search space, we introduce these as what we believe are the most reasonable simplifying assumptions, for practitioners trying to build models that serve K languages relatively equally. The assumption: Developers are likely to build a model that serves K similar languages, and expand down the language list of popularity from there.
>
> These assumptions are of course brittle to different choices. And the formulation could be more powerful if considering token repetition or inter-language transfer, but also we would start hitting the barriers of the amount of data we would need to collect to support learning more variables. And the inter-language variables wouldn’t generalize well even for the hundreds of experiments we ran, because every pair of language synergies needs to be modeled with its own variable.
>
> In short, we propose adding a paragraph to be clearer about the situations this pertains, about the assumptions, and limitations. We do believe these are quite reasonable assumptions given the research question and available compute. We also hope the reviewers recognize these experiments are the furthest down this path that we are aware of.
>
> **Section 2: How would subsets of languages transfer? Does ATLAS capture this?**
>
> We do not capture this, but agree this would be an awesome direction of future work!
>
> **Section 6: Is finetuning continuous pretraining here?**
>
> Yes exactly! We will clarify.
>
> **Conclusion:** added to the draft!
>
> **Finally, we’d like to thank you again for your detailed and constructive feedback. We believe and hope these changes address your core critiques. Please let us know what additional changes we can make to improve the standing of this work in your review. Thank you!**

---

### Meta-Review · Area_Chair_9P3F · 2026-01-07

**Summary:**

While reviewer TADW has questions concerning clarity, the other two reviewers defended the merits of the paper.

I conclude to ignore the score of TADW after reading the authors' response.

**Reviewer Concerns:**

TADW has open questions on clarity, but the authors addressed them.

**Reviewer Scores:**

No changes.

---

### Decision · Program_Chairs · 2026-01-26

Accept (Poster)